# Coupling-based Invertible Neural Networks Are Universal Diffeomorphism Approximators

**Takeshi Teshima**\*
The University of Tokyo, RIKEN
teshima@ms.k.u-tokyo.ac.jp

**Isao Ishikawa**\*
Ehime University, RIKEN
ishikawa.isao.zx@ehime-u.ac.jp

**Koichi Tojo**
RIKEN
koichi.tojo@riken.jp

**Kenta Oono**
The University of Tokyo
kenta_oono@mist.i.u-tokyo.ac.jp

**Masahiro Ikeda**
RIKEN
masahiro.ikeda@riken.jp

**Masashi Sugiyama**
RIKEN, The University of Tokyo
sugi@k.u-tokyo.ac.jp

## Abstract

Invertible neural networks based on coupling flows (CF-INNs) have various machine learning applications such as image synthesis and representation learning. However, their desirable characteristics such as analytic invertibility come at the cost of restricting the functional forms. This poses a question on their representation power: are CF-INNs *universal approximators* for invertible functions? Without a universality, there could be a well-behaved invertible transformation that the CF-INN can never approximate, hence it would render the model class unreliable. We answer this question by showing a convenient criterion: a CF-INN is universal if its layers contain affine coupling and invertible linear functions as special cases. As its corollary, we can affirmatively resolve a previously unsolved problem: whether normalizing flow models based on affine coupling can be *universal distributional approximators*. In the course of proving the universality, we prove a general theorem to show the equivalence of the universality for certain diffeomorphism classes, a theoretical insight that is of interest by itself.

## 1 Introduction

Invertible neural networks based on coupling flows (CF-INNs) are neural network architectures with invertibility by design [1, 2]. Endowed with the analytic-form invertibility and the tractability of the Jacobian, CF-INNs have demonstrated their usefulness in various machine learning tasks such as generative modeling [3–7], probabilistic inference [8–10], solving inverse problems [11], and feature extraction and manipulation [4, 12–14]. The attractive properties of CF-INNs come at the cost of potential restrictions on the set of functions that they can approximate because they rely on carefully designed network layers. To circumvent the potential drawback, a variety of layer designs have been proposed to construct CF-INNs with high representation power, e.g., the affine coupling flow [3, 4, 15–17], the neural autoregressive flow [18–20], and the polynomial flow [21], each demonstrating enhanced empirical performance.

Despite the diversity of layer designs [1, 2], the theoretical understanding of the representation power of CF-INNs has been limited. Indeed, the most basic property as a function approximator, namely the

---

*universal approximation property* (or *universality* for short) [22], has not been elucidated for CF-INNs. The universality can be crucial when CF-INNs are used to learn an invertible transformation (e.g., feature extraction [12] or independent component analysis [14]) because, informally speaking, lack of universality implies that there exists an invertible transformation, even among well-behaved ones, that CF-INN can never approximate, and it would render the model class unreliable for the task of function approximation.

To elucidate the universality of CF-INNs, we first prove a theorem to show the equivalence of the universality for certain diffeomorphism classes, which allows us to reduce the approximation of a general diffeomorphism to that of a much simpler one. By leveraging this problem reduction, we show that CF-INNs based on *affine coupling flows* (ACFs; see Section 2), one of the least expressive flow designs, are in fact universal approximators for a general class of diffeomorphisms. The result can be interpreted as a convenient means to check the universality of a CF-INN: if the flow design can represent ACFs as special cases, then it is universal.

The difficulty in proving the universality of CF-INNs lies in two complications. (1) Only function composition can be leveraged to make complex approximators (e.g., a linear combination is not allowed). We overcome this complication by essentially decomposing a general diffeomorphism into much simpler ones, by using a structural theorem of differential geometry that elucidates the structure of a certain diffeomorphism group. Our equivalence theorem provides a way to take advantage of this technique implicitly. (2) The flow layers tend to be inflexible due to the parametric restrictions. As an extreme example, ACFs can only apply a uniform transformation along the transformed dimension, i.e., the parameter of the transformation cannot depend on the variable which undergoes the transformation. For ACFs, the reduction of the problem allows us to find an approximator with a clear outlook by approximating a step function.

**Our contributions.**    Our contributions are summarized as follows.

1. We present a theorem to show the equivalence of universal approximation properties for certain classes of functions. The result enables the reduction of the task of proving the universality for general diffeomorphisms to that for much simpler coordinate-wise ones.

2. We leverage the result to show that some flow architectures, in particular even ACFs, can be used to construct a CF-INN with the universality for approximating a fairly general class of diffeomorphisms. This result can be seen as a convenient criterion to check the universality of a CF-INN: if the flow designs can reproduce ACF as a special case, it is universal.

3. As a corollary, we give an affirmative answer to a previously unsolved problem, namely the *distributional universality* [18, 21] of ACF-based CF-INNs.

Our result is an interesting application of a deep theorem in differential geometry to investigate the representation power of a neural network architecture.

## 2   Preliminary and goal

In this section, we describe the models analyzed in this study, the notion of universality, and the goal of this paper. We use $\mathbb{R}$ (resp. $\mathbb{N}$) to represent the set of all real numbers (resp. positive integers). For a positive integer $n$, we define $[n]$ as the set $\{1, 2, \ldots, n\}$.

### 2.1   Invertible neural networks based on coupling flows

Throughout the paper, we fix $d \in \mathbb{N}$ and assume $d \geq 2$. For a vector $\boldsymbol{x} \in \mathbb{R}^d$ and $k \in [d-1]$, we define $\boldsymbol{x}_{\leq k}$ as the vector $(x_1, \ldots, x_k)^\top \in \mathbb{R}^k$ and $\boldsymbol{x}_{>k}$ the vector $(x_{k+1}, \ldots, x_d)^\top \in \mathbb{R}^{d-k}$.

**Coupling flows.**    We define a coupling flow (CF) [1] $h_{k,\tau,\theta}$ by $h_{k,\tau,\theta}(\boldsymbol{x}_{\leq k}, \boldsymbol{x}_{>k}) = (\boldsymbol{x}_{\leq k}, \tau(\boldsymbol{x}_{>k}, \theta(\boldsymbol{x}_{\leq k})))$, where $k \in [d-1]$, $\theta \colon \mathbb{R}^k \to \mathbb{R}^l$ and $\tau \colon \mathbb{R}^{d-k} \times \mathbb{R}^l \to \mathbb{R}^{d-k}$ are maps, and $\tau(\cdot, \theta(\boldsymbol{y}))$ is an invertible map for any $\boldsymbol{y} \in \mathbb{R}^k$.

**Affine coupling flows.**    One of the most standard types of CFs is *affine coupling flows* [3, 4, 16, 17]. We define an affine coupling flow $\Psi_{k,s,t} \colon \mathbb{R}^d \to \mathbb{R}^d$ by
$$\Psi_{k,s,t}(\boldsymbol{x}_{\leq k}, \boldsymbol{x}_{>k}) = (\boldsymbol{x}_{\leq k}, \boldsymbol{x}_{>k} \odot \exp(s(\boldsymbol{x}_{\leq k})) + t(\boldsymbol{x}_{\leq k})),$$

where $k \in [d-1]$, $\odot$ is the Hadamard product, $\exp$ is applied in an element-wise manner, and $s, t : \mathbb{R}^k \to \mathbb{R}^{d-k}$ are maps typically parametrized by neural networks.

**Single-coordinate affine coupling flow.** Let $\mathcal{H}$ be a set of functions from $\mathbb{R}^{d-1}$ to $\mathbb{R}$. We define $\mathcal{H}$-*single-coordinate affine coupling flows* by $\mathcal{H}\text{-ACF} := \{\Psi_{d-1,s,t} : s, t \in \mathcal{H}\}$, which is a subclass of ACFs. It is the least expressive flow design appearing in this paper, but we show in Section 3.2 that it can form a CF-INN with universality. We specify the requirements on $\mathcal{H}$ later.

**Invertible linear flows.** We define the set of all affine transforms by $\text{Aff} := \{\boldsymbol{x} \mapsto A\boldsymbol{x} + b : A \in \text{GL}, b \in \mathbb{R}^d\}$, where GL denotes the set of all regular matrices on $\mathbb{R}^d$.

We consider the invertible neural network architectures constructed by composing flow layers:

**Definition 1** (CF-INNs)**.** Let $\mathcal{G}$ be a set consisting of invertible maps. We define the set of invertible neural networks based on $\mathcal{G}$ as

$$\text{INN}_\mathcal{G} := \{W_1 \circ g_1 \circ \cdots \circ W_n \circ g_n : n \in \mathbb{N}, g_i \in \mathcal{G}, W_i \in \text{Aff}\}.$$

When $\mathcal{G}$ can represent the addition of a constant vector, we can obtain the same set of maps by replacing Aff with GL, which has been adopted by previous studies such as Kingma et al. [4]. In fact, it is possible to use only the symmetric group $\mathfrak{S}_d$ that is the permutations of variables, instead of Aff, when $\mathcal{G}$ contains $\mathcal{H}$-ACF. For details, see Appendix H.

## 2.2 Goal: the notions of universality and their relations

Here, we clarify the notion of universality in this paper. First, we prepare some notation. Let $p \in [1, \infty)$ and $m, n \in \mathbb{N}$. For a measurable mapping $f : \mathbb{R}^m \to \mathbb{R}^n$ and a subset $K \subset \mathbb{R}^m$, we define

$$\|f\|_{p,K} := \left( \int_K \|f(x)\|^p \, dx \right)^{1/p},$$

where $\|\cdot\|$ is the Euclidean norm of $\mathbb{R}^n$. We also define $\|f\|_{\sup,K} := \sup_{x \in K} \|f(x)\|$.

**Definition 2** ($L^p$-/sup-universality)**.** Let $\mathcal{M}$ be a model which is a set of measurable mappings from $\mathbb{R}^m$ to $\mathbb{R}^n$. Let $p \in [1, \infty)$, and let $\mathcal{F}$ be a set of measurable mappings $f : U_f \to \mathbb{R}^n$, where $U_f$ is a measurable subset of $\mathbb{R}^m$ which may depend on $f$. We say that $\mathcal{M}$ is an $L^p$-*universal approximator* or *has the $L^p$-universal approximation property* for $\mathcal{F}$ if for any $f \in \mathcal{F}$, any $\varepsilon > 0$, and any compact subset $K \subset U_f$, there exists $g \in \mathcal{M}$ such that $\|f - g\|_{p,K} < \varepsilon$. We define the sup-*universality* analogously by replacing $\|\cdot\|_{p,K}$ with $\|\cdot\|_{\sup,K}$.

We also define the notion of distributional universality. Distributional universality has been used as a notion of theoretical guarantee in the literature of normalizing flows, i.e., probability distribution models constructed using invertible neural networks [2].

**Definition 3** (Distributional universality)**.** Let $\mathcal{M}$ be a model which is a set of measurable mappings from $\mathbb{R}^m$ to $\mathbb{R}^n$. We say that a model $\mathcal{M}$ is a *distributional universal approximator* or *has the distributional universal approximation property* if, for any absolutely continuous[2] probability measure $\mu$ on $\mathbb{R}^m$ and any probability measure $\nu$ on $\mathbb{R}^n$, there exists a sequence $\{g_i\}_{i=1}^\infty \subset \mathcal{M}$ such that $(g_i)_* \mu$ converges to $\nu$ in distribution as $i \to \infty$, where $(g_i)_* \mu := \mu \circ g_i^{-1}$.

If a model $\mathcal{M}$ has the distributional universal approximation property, then it implies $\mathcal{M}$ approximately transforms a known distribution, for example, the uniform distribution on $[0,1]^m$, into any probability measure $\mu$ on $\mathbb{R}^n$, not only absolutely continuous but singular one. There exists another convention that defines the distributional universality as a representation power for only absolutely continuous probability measures. However, since absolutely continuous probability measures are dense in the set of all the probability measures, that convention is equivalent to ours. We include a proof for this fact in Lemma 5 in Appendix A.

The different notions of universality are interrelated. Most importantly, the $L^p$-universality for a certain function class implies the distributional universality (see Lemma 1). Moreover, if a model $\mathcal{M}$ is a sup-universal approximator for $\mathcal{F}$, it is also an $L^p$-universal approximator for $\mathcal{F}$ for any $p \in [1, \infty)$.

**Our goal**   Our goal is to elucidate the representation power of the CF-INNs for some flow architectures $\mathcal{G}$ by proving the $L^p$-universality or sup-universality of $\mathrm{INN}_{\mathcal{G}}$ for a fairly large class of *diffeomorphisms*, i.e., smooth invertible functions. To prove universality, we need to construct a model $g \in \mathrm{INN}_{\mathcal{G}}$ that attains the approximation error $\varepsilon$ for given $f$ and $K$.

## 3   Main results

In this section, we present the main results of this paper on the universality of CF-INNs. The first theorem provides a general proof technique to simplify the problem of approximating diffeomorphisms, and the second theorem builds on the first to show that the CF-INNs based on the affine coupling are $L^p$-universal approximators.

### 3.1   First main result: Equivalence of universal approximation properties

Our first main theorem allows us to lift a universality result for a restricted set of diffeomorphisms to the universality for a fairly general class of diffeomorphisms by showing a certain equivalence of universalities. By using the result to reduce the approximation problem, we can essentially circumvent the major complication in proving the universality of CF-INNs, namely that only function composition can be leveraged to make complex approximators (e.g., a linear combination is not allowed).

First, we define the following classes of invertible functions. Our main theorem later reveals an equivalence of $L^p$-universality/sup-universality for these classes.

**Definition 4** ($C^2$-diffeomorphisms: $\mathcal{D}^2$)**.**  We define $\mathcal{D}^2$ as the set of all $C^2$-diffeomorphisms $f : U_f \to \mathrm{Im}(f) \subset \mathbb{R}^d$ , where $U_f \subset \mathbb{R}^d$ is an open set $C^2$-diffeomorphic to $\mathbb{R}^d$, which may depend on $f$.

**Definition 5** (Triangular transformations: $\mathcal{T}^{\infty}$)**.**  We define $\mathcal{T}^{\infty}$ as the set of all $C^{\infty}$-*increasing triangular* mappings from $\mathbb{R}^d$ to $\mathbb{R}^d$. Here, a mapping $\tau = (\tau_1, \ldots, \tau_d) : \mathbb{R}^d \to \mathbb{R}^d$ is increasing triangular if each $\tau_k(\boldsymbol{x})$ depends only on $\boldsymbol{x}_{\leq k}$ and is strictly increasing with respect to $x_k$.

**Definition 6** (Single-coordinate transformations: $\mathcal{S}_{\mathrm{c}}^r$)**.**  We define $\mathcal{S}_{\mathrm{c}}^r$ as the set of all compactly-supported $C^r$-diffeomorphisms $\tau$ satisfying $\tau(\boldsymbol{x}) = (x_1, \ldots, x_{d-1}, \tau_d(\boldsymbol{x}))$, i.e., those which alter only the last coordinate. In this article, only $r = 0, 2, \infty$ appear, and we mainly focus on $\mathcal{S}_{\mathrm{c}}^{\infty} (\subset \mathcal{T}^{\infty})$. Here, a bijection $\tau : \mathbb{R}^d \to \mathbb{R}^d$ is compactly supported if $\tau = \mathrm{Id}$ outside some compact set.

Among the above classes of invertible functions, $\mathcal{D}^2$ is our main approximation target, and it is a fairly large class: it contains any $C^2$-diffeomorphism defined on the entire $\mathbb{R}^d$, an open convex set, or more generally a star-shaped open set. The class $\mathcal{T}^{\infty}$ relates to the distributional universality as we will see in Lemma 1. The class $\mathcal{S}_{\mathrm{c}}^{\infty}$ is a much simpler class of diffeomorphisms that we use as a stepladder for showing the universality for $\mathcal{D}^2$.

Now we are ready to state the first main theorem. It reveals an equivalence among the universalities for $\mathcal{D}^2$, $\mathcal{T}^{\infty}$, and $\mathcal{S}_{\mathrm{c}}^{\infty}$, under mild regularity conditions. We can use the theorem to lift up the universality for $\mathcal{S}_{\mathrm{c}}^{\infty}$ to that for $\mathcal{D}^2$.

**Theorem 1** (Equivalence of Universality)**.**  *Let $p \in [1, \infty)$ and let $\mathcal{G}$ be a set of invertible functions.*

*(A) If all elements of $\mathcal{G}$ are piecewise $C^1$-diffeomorphisms, then the $L^p$-universal approximation properties of $\mathrm{INN}_{\mathcal{G}}$ for $\mathcal{D}^2$, $\mathcal{T}^{\infty}$ and $\mathcal{S}_{\mathrm{c}}^{\infty}$ are all equivalent.*

*(B) If all elements of $\mathcal{G}$ are locally bounded, then the* sup-*universal approximation properties of $\mathrm{INN}_{\mathcal{G}}$ for $\mathcal{D}^2$, $\mathcal{T}^{\infty}$ and $\mathcal{S}_{\mathrm{c}}^{\infty}$ are all equivalent.*

The proof is provided in Appendix B. For the definitions of the piecewise $C^1$-diffeomorphisms and the locally bounded maps, see Appendix E. The regularity conditions in (A) and (B) assure that function composition within $\mathcal{G}$ is compatible with approximations (see Appendix F for details), and they are usually satisfied, e.g., continuous maps are locally bounded.

If one of the two universality properties in Theorem 1 is satisfied, the model is also a distributional universal approximator. Let $p \in [1, \infty)$, and we have the following.

**Lemma 1.**  *An $L^p$-universal approximator for $\mathcal{T}^{\infty}$ is a distributional universal approximator.*

Table 1: CF-INN instances analyzed in this work (*Model*: the considered CF-INN architecture. *Flow type*: the flow layer architecture. *Universality (this)*: the universal approximation property that this work has shown. *Universality (prev.)*: previously claimed universal approximation property. ) Our proof techniques are easy to apply to analyze the universality of various CF-INN architectures.

| Model | Flow type | Universality (this) | Universality (prev.) |
|---|---|---|---|
| $\text{INN}_{\mathcal{H}\text{-ACF}}$ | Affine coupling [3, 4, 16, 17] | $L^p$-universal | - |
| $\text{INN}_{\text{DSF}}$ | Deep sigmoidal flow [18] | sup-universal | Distributional [18] |
| $\text{INN}_{\text{SoS}}$ | Sum-of-squares polynomial flow [21] | sup-universal | Distributional [21] |

Since sup-universality implies $L^p$-universality, Lemma 1 can be combined with both cases of (A) and (B) in Theorem 1. The proof is based on the existence of a triangular map connecting two absolutely continuous distributions [23]. See Appendix A for details. Note that the previous studies [18, 21] have discussed the distributional universality of some flow architectures essentially via showing the sup-universality for $\mathcal{T}^\infty$. Lemma 1 clarifies that the weaker notion of $L^p$-universality is sufficient for the distributional universality, which can also apply to the case (A) in Theorem 1.

**Application to previously proposed CF-INN architectures.** Theorem 1 can upgrade a previously known sup-universality for $\mathcal{T}^\infty$ of a CF-INN architecture to that for $\mathcal{D}^2$. As examples, *deep sigmoidal flows* (DSF; a version of neural autoregressive flows [18]) and *sum-of-squares polynomial flows* (SoS; [21]) can both yield CF-INNs with the sup-universal approximation property for $\mathcal{D}^2$. We provide the proof in Appendix G. See Table 1 for a summary of the results. See Section 5.1 for a comparison with previous theoretical analyses on normalizing flows.

## 3.2 Second main result: $L^p$-universal approximation property of $\text{INN}_{\mathcal{H}\text{-ACF}}$

Our second main theorem reveals the $L^p$-universality of $\text{INN}_{\mathcal{H}\text{-ACF}}$ for $\mathcal{S}_c^0$ (hence for $\mathcal{S}_c^\infty$), which can be combined with Theorem 1 to show its $L^p$-universality for $\mathcal{D}^2$. We define $C_c^\infty(\mathbb{R}^{d-1})$ as the set of all compactly-supported $C^\infty$ maps from $\mathbb{R}^{d-1}$ to $\mathbb{R}$.

**Theorem 2** ($L^p$-universality of $\text{INN}_{\mathcal{H}\text{-ACF}}$). *Let $p \in [1, \infty)$. Assume $\mathcal{H}$ is a sup-universal approximator for $C_c^\infty(\mathbb{R}^{d-1})$ and that it consists of piecewise $C^1$-functions. Then, $\text{INN}_{\mathcal{H}\text{-ACF}}$ is an $L^p$-universal approximator for $\mathcal{S}_c^0$.*

We provide a proof in Appendix D. For the definition of piecewise $C^1$-functions, see Appendix E. Theorem 2 can be combined with Theorem 1 to show that $\text{INN}_{\mathcal{H}\text{-ACF}}$ is an $L^p$-universal approximator for $\mathcal{D}^2$. Examples of $\mathcal{H}$ satisfying the condition of Theorem 2 include multi-layer perceptron models with the *rectifier linear unit* (ReLU) activation [24] and a linear-in-parameter model with smooth universal kernels [25]. The result can be interpreted as a convenient criterion to check the universality of a CF-INN: if the flow architecture $\mathcal{G}$ contains ACFs (or even just $\mathcal{H}$-ACF with sufficiently expressive $\mathcal{H}$) as special cases, then $\text{INN}_\mathcal{G}$ is an $L^p$-universal approximator for $\mathcal{D}^2$.

By combining Theorem 1, Theorem 2, and Lemma 1, we can affirmatively answer a previously unsolved problem [1, p.13]: the distributional universality of CF-INN based on ACFs.

**Theorem 3** (Distributional universality of $\text{INN}_{\mathcal{H}\text{-ACF}}$). *Under the conditions of Theorem 2, $\text{INN}_{\mathcal{H}\text{-ACF}}$ is a distributional universal approximator.*

**Implications of Theorem 2 and Theorem 3.** Theorem 2 implies that, if $\mathcal{G}$ contains $\mathcal{H}$-ACF as special cases, then $\text{INN}_\mathcal{G}$ is an $L^p$-universal approximator for $\mathcal{D}^2$. In light of Theorem 3, it is also a distributional universal approximator, hence we can confirm the theoretical plausibility for using it for normalizing flows. Such examples of $\mathcal{G}$ include the *nonlinear squared flow* [26], *Flow++* [20], the *neural autoregressive flow* [18], and the *sum-of-squares polynomial flow* [21]. The result may not immediately apply to the typical *Glow* [4] models for image data that use the 1x1 invertible convolution layers and convolutional neuralnetworks for the coupling layers. However, the Glow architecture for non-image data [11, 14] can be interpreted as $\text{INN}_\mathcal{G}$ with ACF layers, hence it is both an $L^p$-universal approximator for $\mathcal{D}^2$ and a distributional universal approximator.

# 4 Proof outline

In this section, we outline the proof ideas of our main theorems to provide an intuition for the constructed approximator and derive reusable insight for future theoretical analyses.

## 4.1 Proof outline for Theorem 1

Here, we outline the equivalence proof of Theorem 1. For details, see Appendix B. Since we have $\mathcal{S}_c^\infty \subset \mathcal{T}^\infty \subset \mathcal{D}^2$, it is sufficient to prove that the universal approximation properties for $S^\infty$ implies that for $\mathcal{D}^2$. Note that the proofs do not change for $L^p$-universality and sup-universality.

Therefore, we focus on describing the reduction from $\mathcal{D}^2$ to $\mathcal{S}_c^\infty$. Since the approximation of $\mathcal{S}_c^2$ can be reduced to that of $\mathcal{S}_c^\infty$ by a standard mollification argument (see Appendix B.2), we show a reduction from $\mathcal{D}^2$ to $\mathcal{S}_c^2$:

**Theorem 4.** *For any element $f \in \mathcal{D}^2$ and compact subset $K \subset U_f$, there exist $n \in \mathbb{N}$, $W_1, \ldots, W_n \in$ Aff, and $\tau_1, \ldots, \tau_n \in \mathcal{S}_c^2$ such that $f(x) = W_1 \circ \tau_1 \circ \cdots \circ W_n \circ \tau_n(x)$ for all $x \in K$.*

Behind the scenes, Theorem 4 reduces $\mathcal{D}^2$ to $\mathcal{S}_c^2$ in four steps:

$$\mathcal{D}^2 \rightsquigarrow \mathrm{Diff}_c^2 \rightsquigarrow \text{Flow endpoints} \rightsquigarrow \text{nearly-Id} \rightsquigarrow \mathcal{S}_c^2$$

Here, $A \rightsquigarrow B$ ($A$ *is reduced to* $B$) indicates that the universality for $A$ follows from that for $B$, and Id denotes the identity map. We explain each reduction step in the below.

*From $\mathcal{D}^2$ to $\mathrm{Diff}_c^2$.* We consider a special subset $\mathrm{Diff}_c^2 \subset \mathcal{D}^2$, which is the group of *compactly-supported* $C^2$-diffeomorphisms on $\mathbb{R}^d$ whose group operation is functional composition. Here, a bijection $f : \mathbb{R}^d \to \mathbb{R}^d$ is compactly supported if $f = \mathrm{Id}$ outside some compact set. Proposition 1 below reduces the problem of the universality for $\mathcal{D}^2$ to that for $\mathrm{Diff}_c^2$.

**Proposition 1.** *For any $f \in \mathcal{D}^2$ and any compact subset $K \subset U_f$, there exist $h \in \mathrm{Diff}_c^2$, $W \in$ Aff, such that for all $\boldsymbol{x} \in K$, $f(\boldsymbol{x}) = W \circ h(\boldsymbol{x})$.*

*From $\mathrm{Diff}_c^2$ to flow endpoints.* In order to construct an approximation for the elements of $\mathcal{D}^2$, we devise its subset that we call the *flow endpoints*. A flow endpoint is an element of $\mathrm{Diff}_c^2$ which can be represented as $\phi(1)$ using an "additive" continuous map $\phi : [0, 1] \to \mathrm{Diff}_c^2$ with $\phi(0) = \mathrm{Id}$. Here, "additivity" means $\phi(s) \circ \phi(t) = \phi(s + t)$ for any $s, t \in [0, 1]$ with $s + t \in [0, 1]$. This additivity will be later used to decompose a flow endpoint into a composition of some mildly-behaved fragments of the flow map. Note that we equip $\mathrm{Diff}_c^2$ with the Whitney topology [27, Proposition 1.7.(9)] to define the continuity of the map $\phi$. The importance of the flow endpoints lies in the following lemma that we prove in Appendix C:

**Lemma 2.** *Any element in $\mathrm{Diff}_c^2$ can be represented as a finite composition of flow endpoints.*

Lemma 2 is essentially due to Fact 1, which is the following structure theorem in differential geometry attributed to Herman, Thurston [28], Epstein [29], and Mather [30, 31]:

**Fact 1.** *The group $\mathrm{Diff}_c^2$ is simple, i.e., any normal subgroup $H \subset \mathrm{Diff}_c^2$ is either $\{\mathrm{Id}\}$ or $\mathrm{Diff}_c^2$.*

*From flow endpoints to nearly-Id.* The flow endpoints in $\mathrm{Diff}_c^2$ can be decomposed into "nearly-Id" elements in $\mathrm{Diff}_c^2$ by leveraging its additivity property, as in the following proposition. Let $\|\cdot\|_{\mathrm{op}}$ denote the operator norm.

**Proposition 2.** *For any $f \in \mathrm{Diff}_c^2$, there exist finite elements $g_1, \ldots, g_r \in \mathrm{Diff}_c^2$ such that $f = g_1 \circ \cdots \circ g_r$ and $\sup_{x \in \mathbb{R}^d} \|Dg_i(x) - I\|_{\mathrm{op}} < 1$, where $Dg_i$ is the Jacobian of $g_i$.*

Proposition 2 leverages the continuity of the flows with respect to the Whitney topology of $\mathrm{Diff}_c^2$: $\phi(1/n)$ uniformly converges to the identity map both in its values and *its Jacobian* when $n \to \infty$. Thus, any flow endpoint $\phi(1)$ can be represented by an $n$-time composition of $\phi(1/n)$ each of which is close to identity (nearly-Id) when $n$ is sufficiently large.

*From nearly-Id to $\mathcal{S}_c^2$.* The nearly-Id elements, $g \in \mathrm{Diff}_c^2$ in Proposition 2, can be decomposed into elements of $\mathcal{S}_c^2$ and permutation matrices:

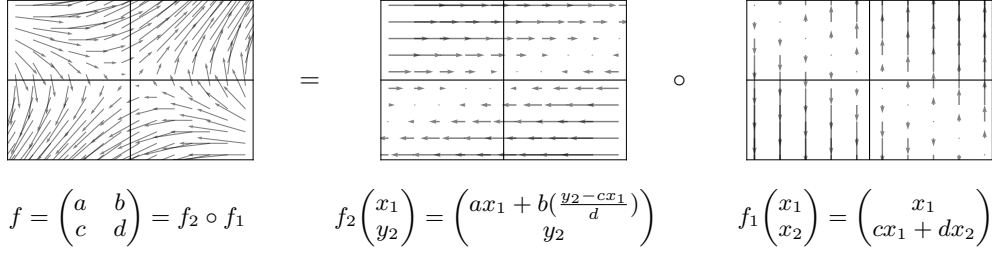

$$f = \begin{pmatrix} a & b \\ c & d \end{pmatrix} = f_2 \circ f_1 \qquad f_2\begin{pmatrix} x_1 \\ y_2 \end{pmatrix} = \begin{pmatrix} ax_1 + b(\frac{y_2 - cx_1}{d}) \\ y_2 \end{pmatrix} \qquad f_1\begin{pmatrix} x_1 \\ x_2 \end{pmatrix} = \begin{pmatrix} x_1 \\ cx_1 + dx_2 \end{pmatrix}$$

Figure 1: A nearly-Id transformation $f$ can be decomposed into coordinate-wise ones ($f_1$ and $f_2$: realized by $\mathcal{S}_c^2$ and permutations). The arrows indicate the transportation of the positions. A general nonlinear $f$ can be analogously decomposed by Proposition 3 when $f$ satisfies certain conditions.

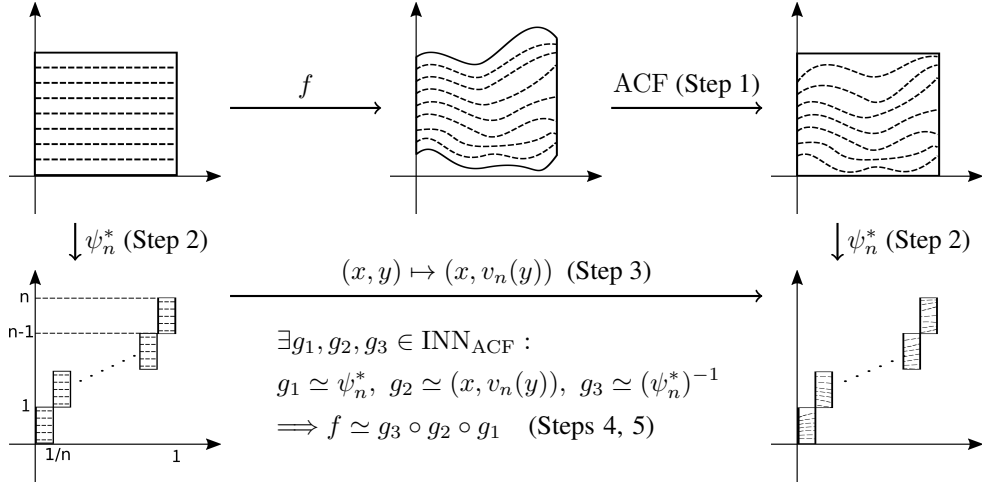

Figure 2: Illustration of the proof technique for the $L^p$-universal approximation property of $\mathrm{INN}_{\mathrm{ACF}}$ for $\mathcal{S}_c^0$. The symbol $\simeq$ indicates approximation to arbitrary precision.

**Proposition 3.** *For any* $g \in \mathrm{Diff}_c^2$ *with* $\sup_{x \in \mathbb{R}^d} \|Dg(x) - I\|_{\mathrm{op}} < 1$, *there exist $d$ elements* $\tau_1, \ldots, \tau_d \in \mathcal{S}_c^2$ *and permutation matrices* $\sigma_1, \ldots, \sigma_d$ *such that*

$$g = \sigma_1 \circ \tau_1 \circ \cdots \circ \sigma_d \circ \tau_d.$$

The machinery of this decomposition is illustrated in Figure 1.

### 4.2 Proof outline for Theorem 2

Here, we give the proof outline of Theorem 2. For details, see Appendix D. The main difficulty in constructing the approximator is the restricted functional form of ACFs. However, the problem reduction by Theorem 1 allows us to construct an approximator by approximating a step function.

For illustration, we only describe the case for $d = 2$ and $K \subset [0,1]^2$. For complete proof of Theorem 2, see Appendix D. Let $f(x, y) = (x, u(x, y))$ be the target function, where $u(\cdot, y)$ is a continuous function that is strictly increasing for each $y$ (i.e., $f \in \mathcal{S}_c^0$). For the compact set $K \subset [0,1]^2 \subset \mathbb{R}^2$, we find $g \in \mathrm{INN}_{\mathcal{H}\text{-}\mathrm{ACF}}$ arbitrarily approximating $f$ on $K$ as follows (Figure 2).

Step 1. **Align the image into the square:** First, without loss of generality, we may assume that the image $f([0,1]^2)$ is again $[0,1]^2$. Indeed, we can align the image so that $u(x,1) = 1$ and $u(x,0) = 0$ for all $x \in [0,1]$ by using only an ACF $\Psi_{1,s,t}$ with continuous $s$ and $t$, which can be approximated by $\mathcal{H}\text{-}\mathrm{ACF}$.

Step 2. **Slice the squares and stagger the pieces:** We consider an imaginary ACF $\psi_n^* := \Psi_{1,1,t_n}$ defined using a discontinuous step function $t_n := \sum_{k=0}^{n} k \mathbf{1}_{[k/n,(k+1)/n)}$. The map $\psi_n^*$ splits $[0,1]^2$ into pieces and staggers them so that a coordinate-wise independent transformation (e.g., $v_n$ in Step 3), which is uniform along the $x$-axis, can affect each piece separately.

Step 3. **Express $f$ by a coordinate-wise independent transformation:** We construct a continuous increasing function $v_n : \mathbb{R} \to \mathbb{R}$ such that for $y \in [k, k+1)$, $v_n(y) = u(k/n, y) + k$ ($k = 0, \ldots, n-1$). A direct computation shows that $\tilde{f}_n := (\psi_n^*)^{-1} \circ (\cdot, v_n(\cdot)) \circ \psi_n^*$ arbitrarily approximates $f$ on $[0,1]^2$ if we increase $n$. We take a sufficiently large $n$.

Step 4. **Approximate the coordinate-wise independent transformation $v_n$:** We find an element of $\mathrm{INN}_{\mathcal{H}\text{-ACF}}$ sufficiently approximating $(\cdot, v_n(\cdot))$ on $[0,1] \times [0,n]$. This is realized based on a lemma that we can construct an approximator for any element of $\mathcal{S}_c^0$ of the form $(x,y) \mapsto (x, v(y))$ on any compact set in $\mathbb{R}^2$.

Step 5. **Approximate $\psi_n^*$ and combine the approximated constituents to approximate $\tilde{f}_n$:** We can also approximate $\psi_n^*$ and its inverse by ACFs based on the universality of $\mathcal{H}$. Finally, composing the approximated constituents gives an approximation of $f$ on $[0,1]^2$ with arbitrary precision (see Appendix F).

# 5 Related work and discussions

In this section, we relate the contribution of this work to the literature on the representation power of invertible neural networks.

## 5.1 Relation to previous theoretical analyses for normalizing flow models

The distributional universality of *normalizing flows* constructed using CF-INNs has been addressed in previous studies such as [18, 21]. Previously proposed architectures with distributional universality include the neural autoregressive flows [18] and the sum-of-squares polynomial flows [21]. Our findings elucidate the much stronger universalities of these architectures, namely the sup-universality for $\mathcal{D}^2$, which enhances the reliability of these models in the tasks where function approximation rather than distribution approximation is crucial, e.g., feature extraction [12, 14]. The *deeply lazy maps (DLMs)* proposed in Brennan et al. [32] can also be considered as a class of CF-INNs. Brennan et al. [32] provided a sufficient condition for a series of DLMs to result in some normalizing flows that weakly converge to a target distribution.

Huang et al. [33] has also shown that a general flow architecture realized by arbitrary *autoregressive neural networks* is a universal distributional approximator. Although Proposition 1 of Huang et al. [33] was formulated to analyze the *inverse autoregressive flow (IAF)* [17], which can be regarded as a composition of ACFs, it should be noted that the analyzed architecture is the class of arbitrary autoregressive neural networks, hence it does not provide a guarantee for the IAF in Kingma et al. [17]. In this regard, Theorem 3 is the first to show the distributional universality of ACF-based CF-INNs to the best of our knowledge.

## 5.2 Theoretical guarantee for other invertible neural network architectures

**One-dimensional case.** In the one-dimensional case ($d = 1$), strict monotonicity is a necessary and sufficient condition for a function to be invertible. In this case, there have been a few invertible neural network architectures with sup-universality for the set of all homeomorphisms on $\mathbb{R}$, e.g., *monotonic networks* [34] and *rational quadratic splines* [35]. These models complement CF-INNs in that they provide an invertible neural network only in the one-dimensional case, whereas the latter can be defined only in the multi-dimensional case.

**Limited approximation efficiency of residual-flow based normalizing flows.** Kong et al. [36] provided a quantitative theoretical analysis of the representation power of normalizing flows constructed by using INNs based on *residual flows*, which is another approach for designing a flow layer. Specifically, it presented a lower bound on the number of layers required for approximating a certain distribution using previously proposed residual-flow based normalizing flows. The result, albeit for a different type of flow layers from CFs, shows the importance of developing flow layers with improved

approximation efficiency. In such an endeavor, our results can provide a simple route to confirming the universality of those CFs to be designed in the future for improved approximation efficiency.

**Relation to examples of functions that cannot be approximated by NODEs.** Neural ordinary differential equations (NODEs) [37, 38] can be considered as another design of invertible flow layers different from CFs. Zhang et al. [39] formulated its Theorem 1 to show that NODEs are not universal approximators by presenting a function that a NODE cannot approximate. The existence of this counterexample does not contradict our result because our approximation target $\mathcal{D}^2$ is different from the function class considered in Zhang et al. [39]: the class in Zhang et al. [39] can contain discontinuous maps whereas the elements of $\mathcal{D}^2$ are smooth and invertible. Also, in Proposition 1, we cap an affine transformation (realizable by $\mathrm{INN}_{\mathcal{G}}$) on top of the target function to reduce the approximation of $\mathcal{D}^2$ to that of $\mathrm{Diff}_c^2$. Such an affine transformation may enhance the approximation capacity by allowing a certain set of transformations, e.g., coordinate-wise sign flipping.

### 5.3 The strength of the representation power of $\mathrm{INN}_{\mathcal{H}\text{-ACF}}$

In this study, we showed the $L^p$-universal approximation property of $\mathrm{INN}_{\mathcal{H}\text{-ACF}}$. While the $L^p$-universality is likely to suffice for developing probabilistic risk bounds for machine learning tasks [40, 41] and for showing distributional universality, whether $\mathrm{INN}_{\mathcal{H}\text{-ACF}}$ is a sup-universal approximator for $\mathcal{D}^2$ remains an open question. Our conjecture is negative due to the following theoretical observation. The sup-universality requires a precise approximation uniformly everywhere while the $L^p$-universality can allow an approximation error on negligible regions. As described in Section 4.2, we used a smooth approximation of step functions to show the $L^p$-universality of $\mathrm{INN}_{\mathcal{H}\text{-ACF}}$. Intuitively, approximating the step functions and composing them can accumulate errors around the discontinuity points, so that it can retain the $L^p$-universality but it can affect the sup-universality. Since the step functions are devised to bypass the uniformity of the transformation by ACFs, we conjecture that the difficulty is intrinsic and a sup-universality is unlikely to hold for $\mathrm{INN}_{\mathcal{H}\text{-ACF}}$.

## 6 Conclusion

In this study, we elucidated the representation power of CF-INNs by proving their $L^p$-universality or sup-universality for $\mathcal{D}^2$. Along the course, we invoked a structure theorem from differential geometry to establish an equivalence of the universalities for $\mathcal{D}^2$, $\mathcal{S}_c^\infty$, and $\mathcal{T}^\infty$, which itself is of theoretical interest. Our result advances the theoretical understanding of CF-INNs by formally showing that most of the CF-INN architectures already yield $L^p$-universal approximators and that the different flow layer designs purely contribute to the efficiency of approximation, not much to the capacity of the model class. Comparing the approximation efficiency of different layer designs is an important area in future work. Also, the approximation efficiency for a better-behaved subset of $\mathcal{D}^2$ (e.g., bi-Lipschitz ones) remains as an open question for future research.

## Broader Impact

This work advances the theoretical understanding of *invertible neural networks* (INNs), a recently emerging function model in machine learning. Since the major contribution of this paper is to provide a framework to theoretically guarantee the representation power of INNs, the presented results are likely to promote the use of INNs in various machine learning tasks, although an immediate direct impact on the practice of machine learning is unlikely.

## Acknowledgments and Disclosure of Funding

The authors would like to thank the anonymous reviewers for the insightful discussions. We would also like to thank Dr. Taiji Suzuki, Associate Professor of the University of Tokyo, for his valuable comments and fruitful discussions on the distributional universality. This work was supported by RIKEN Junior Research Associate Program. TT was supported by Masason Foundation. II and MI were supported by CREST:JPMJCR1913. MS was supported by KAKENHI 20H04206.

## Footnotes

[2]In this paper, we say a measure on the Euclidean space is *absolutely continuous* when it is absolutely continuous with respect to the Lebesgue measure.

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
