[Supplementary Material]

# Appendices

This is the Supplementary Material for "Coupling-based Invertible Neural Networks Are Universal Diffeomorphism Approximators." Table 2 summarizes the abbreviations and the symbols used in the paper. Figure 3 depicts the relations among the notions of universalities appearing in this paper and how they are connected by the sections in this Supplementary Material.

Table 2: Abbreviation and notation table

| Abbreviation/Notation | Meaning |
|---|---|
| CF-INN | Invertible neural networks based on coupling flow |
| IAF | Inverse autoregressive flow |
| DSF | Deep sigmoidal flow |
| SoS | Sum-of-squares polynomial flow |
| MLP | Multi-layer perceptron |
| CF, $h_{k,\tau,\theta}$ | Coupling flow |
| ACF, $\Psi_{k,s,t}$ | Affine coupling flow |
| $\mathcal{H}$ | Set of functions from $\mathbb{R}^{d-1}$ to $\mathbb{R}$ |
| $\mathcal{H}$-ACF, $\Psi_{d-1,s,t}$ | $\mathcal{H}$-single-coordinate affine coupling flows $(s, t \in \mathcal{H})$ |
| Aff | Set of invertible affine transformations |
| GL | Set of invertible linear transformations |
| $\mathcal{G}$ | Generic notation for a set of invertible functions |
| $\mathrm{INN}_{\mathcal{G}}$ | Set of invertible neural networks based on $\mathcal{G}$ |
| $\mathcal{D}^2$ | Set of all $C^2$-diffeomorphisms with $C^2$-diffeomorphic domains |
| $\mathcal{T}^\infty$ | Set of all $C^\infty$-increasing triangular mappings |
| $\mathcal{S}_{\mathrm{c}}^r$ | Set of all $C^r$-single-coordinate transformations |
| $\mathrm{Diff}_{\mathrm{c}}^2$ | Group of compactly-supported $C^2$-diffeomorphisms (on $\mathbb{R}^d$) |
| $\|\cdot\|$ | Euclidean norm |
| $\|\cdot\|_{\mathrm{op}}$ | Operator norm |
| $\|\cdot\|_{p,K}$ | $L^p$-norm $(p \in [1, \infty))$ on a subset $K \subset \mathbb{R}^d$ |
| $\|\cdot\|_{\mathrm{sup},K}$ | Supremum norm on a subset $K \subset \mathbb{R}^d$ |
| $\mathbf{1}_A(\cdot)$ | Indicator (characteristic) function of $A$ |

## A  Proof of Lemma 1: From $L^p$-universality to distributional universality

Here, we prove Lemma 3, which corresponds to Lemma 1 in the main text.

First, note that the larger $p$, the stronger the notion of $L^p$-universality: if a model $\mathcal{M}$ is an $L^p$-universal approximator for $\mathcal{F}$, it is also an $L^q$-universal approximator for $\mathcal{F}$ for all $1 \leq q \leq p$. In particular, we use this fact with $q = 1$ in the following proof.

**Lemma 3** (Lemma 1 in the main text). *Let $p \in [1, \infty)$. Suppose $\mathcal{M}$ is an $L^p$-universal approximator for $\mathcal{T}^\infty$. Then $\mathcal{M}$ is a distributional universal approximator.*

*Proof.* We denote by $\mathrm{BL}_1$ the set of bounded Lipschitz functions $f: \mathbb{R}^d \to \mathbb{R}$ satisfying $\|f\|_{\mathrm{sup},\mathbb{R}^d} + L_f \leq 1$, where $L_f$ denotes the Lipschitz constant of $f$. Let $\mu, \nu$ be absolutely continuous probability measures, and take any $\varepsilon > 0$. By Theorem 11.3.3 in [42], it suffices to show that there exists $g \in \mathcal{M}$ such that

$$\beta(g_*\mu, \nu) := \sup_{f \in \mathrm{BL}_1} \left| \int_{\mathbb{R}^d} f \, dg_*\mu - f \, d\nu \right| < \varepsilon.$$

Let $p, q \in L^1(\mathbb{R}^d)$ be the density functions of $\mu$ and $\nu$ respectively. Let $\phi \in L^1(\mathbb{R}^d)$ be a positive $C^\infty$-function such that $\int_{\mathbb{R}^d} \phi(x)dx = 1$ (for example, Gaussian distribution), and for $t > 0$, put $\phi_t(x) := t^{-d}\phi(x/t)$. We define $\mu_t := \phi_t * p dx$ and $\nu_t := \phi_t * q dx$. Since both $\|\phi_t * p - p\|_{1,\mathbb{R}^d}$ and

Figure 3: Informal diagram of the relations among propositions and lemmas connecting them. Here, $p \in [1, \infty)$. *S.C.* stands for "special case" and indicates that the notion of universality implies the other as a special case. *DSF* stands for *deep sigmoidal flow*, and *SoS* stands for *sum-of-squares polynomial flow*.

$\|\phi_t * q - q\|_{1, \mathbb{R}^d}$ converges to 0 as $t \to 0$, there exists $t_0 > 0$ such that for any continuous mapping $G : \mathbb{R}^d \to \mathbb{R}^d$,

$$\left| \int_{\mathbb{R}^d} f \, dG_* \mu_{t_0} - f \, dG_* \mu \right| < \frac{\|f\|_{\sup, \mathbb{R}^d} \, \varepsilon}{5}, \quad \left| \int_{\mathbb{R}^d} f \, d\nu_{t_0} - f \, d\nu \right| < \frac{\|f\|_{\sup, \mathbb{R}^d} \, \varepsilon}{5}.$$

By using Lemma 4 below, there exists $T \in \mathcal{T}^\infty$ such that $T_* \mu_{t_0} = \nu_{t_0}$. Let $K \subset \mathbb{R}^d$ be a compact subset such that

$$1 - \mu_{t_0}(K) < \frac{\varepsilon}{5}.$$

By the assumption, there exists $g \in \mathcal{M}$ such that

$$\int_K |T(x) - g(x)| dx < \frac{\varepsilon}{5 \sup_{x \in K} |\phi_{t_0} * p(x)|}.$$

Thus for any $f \in \mathrm{BL}_1$, we have

$$\left| \int_{\mathbb{R}^d} f \, dg_*\mu - f \, d\nu \right|$$

$$\leq \left| \int_{\mathbb{R}^d} f \, dg_*\mu_{t_0} - f \, dg_*\mu \right| + \left| \int_{\mathbb{R}^d} f \, d\nu_{t_0} - f \, d\nu \right|$$

$$+ \left| \int_{\mathbb{R}^d \setminus K} f \circ T \, d\mu_{t_0} \right| + \left| \int_{\mathbb{R}^d \setminus K} f \circ g \, d\mu_{t_0} \right| + \int_K |f(T(x)) - f(g(x))| \, d\mu_{t_0}(x)$$

$$< \frac{\|f\|_{\mathrm{sup},\mathbb{R}^d} \, \varepsilon}{5} + \frac{\|f\|_{\mathrm{sup},\mathbb{R}^d}}{5} + \frac{\|f\|_{\mathrm{sup},\mathbb{R}^d} \, \varepsilon}{5} + \frac{\|f\|_{\mathrm{sup},\mathbb{R}^d} \, \varepsilon}{5} + \frac{L_f \varepsilon}{5}$$

$$\leq \varepsilon,$$

where $L_f$ is the Lipschitz constant of $f$. Here we used $\|f\|_{\mathrm{sup},\mathbb{R}^d} + L_f \leq 1$. Therefore, we have $\beta(g_*\mu, \nu) < \varepsilon$. $\qquad\square$

The following lemma is essentially due to [43].

**Lemma 4.** *Let $\mu$ be a probability measure on $\mathbb{R}^d$ with a $C^\infty$ density function $p$. Let $U := \{x \in \mathbb{R}^d : p(x) > 0\}$. Then there exists a diffeomorphism $T : U \to (0,1)^d$ such that its Jacobian is upper triangular matrix with positive diagonal, and $T_*\mu = \mathrm{U}(0,1)^d$. Here, $\mathrm{U}(0,1)^d$ is the uniform distribution on $[0,1]^d$.*

*Proof.* Let $q_i(x_1, \ldots, x_i) := \int_{\mathbb{R}^{d-i}} p(x_1, \ldots, x_{i+1}, \ldots, x_d) \, dx_{i+1} \ldots dx_d$. Then we define $T : U \to (0,1)^d$ by

$$T(x_1, \ldots, x_d) := \left( \int_{-\infty}^{x_i} \frac{q_i(x_1, \ldots, x_{i-1}, y)}{q_{i-1}(x_1, \ldots, x_{i-1})} dy \right)_i.$$

Then we see that $T$ is a diffeomorphism and its Jacobian is upper triangular with positive diagonal elements. Moreover, by a direct computation, we have $T_* d\mu = U(0,1)$. $\qquad\square$

We include a proof for the statement that that any probability measure on $\mathbb{R}^m$ is arbitrarily approximated by an absolutely continuous probability measure in the weak convergence topology:

**Lemma 5.** *Let $\mu$ be a arbitrary probability measure of $\mathbb{R}^m$. Then there exists a sequence $\{\mu_n\}_{n=1}^\infty$ such that $\mu_n$ weakly converges to $\mu$.*

*Proof.* Let $\phi$ be a positive $C^\infty$ function such that $\int_{\mathbb{R}^m} \phi(x) dx = 1$. For $t > 0$, put $\phi_t(x) := t^{-m}\phi(x/t)$. We define

$$w_t(x) = \int_{\mathbb{R}^m} \phi_t(x - y) d\mu(y).$$

We prove the absolutely continuous measure $w_t dx$ weakly converges to $\mu$ as $t \to 0$. In fact, for any bounded continuous function $f$, we have

$$\left| \int_{\mathbb{R}^m} f w_t dx - \int f d\mu \right| = \left| \int \int_{\mathbb{R}^m} f(y + tx) - f(y)\phi(x) dx d\mu(y) \right|$$

$$\leq \int \int_{\mathbb{R}^m} |f(y + tx) - f(y)|\phi(x) dx d\mu(y).$$

Since $f$ is bounded and $\phi$ is absolutely integrable, by the dominated convergence theorem, as $t \to 0$, we have

$$\int_{\mathbb{R}^m} f w_t dx \to \int f d\mu,$$

namely, $w_t dx$ weakly converges to $\mu$. $\qquad\square$

# B   Proof of Theorem 1: Equivalence of universality properties

In this section, we provide the proof details of Theorem 1 in the main text. Section B.1 explains the reduction from $\mathcal{D}^2$ to $\mathrm{Diff}_{\mathrm{c}}^2$, and Section B.2 explains the reduction from $\mathrm{Diff}_{\mathrm{c}}^2$ to $\mathcal{S}_{\mathrm{c}}^\infty$ and permutations of variables.

Here, we formally repost the proof of Theorem 1 which has been essentially completed in Section 4.1.

*Proof of Theorem 1.* Since we have $\mathcal{S}_{\mathrm{c}}^\infty \subset \mathcal{T}^\infty \subset \mathcal{D}^2$, it is sufficient to prove that the universal approximation properties for $S^\infty$ imply those for $\mathcal{D}^2$. Therefore, we focus on describing the reduction from $\mathcal{D}^2$ to $\mathcal{S}_{\mathrm{c}}^\infty$. First, by combining Lemma 11 with the $L^p$-universality (in the case A) or the sup-universality (in the case B) of $\mathrm{INN}_{\mathcal{G}}$ for $\mathcal{S}_{\mathrm{c}}^\infty$, we obtain the $L^p$-universal (resp. sup-universal) approximation property for $\mathcal{S}_{\mathrm{c}}^2$. Now, in light of Lemma 6 and Theorem 5, we obtain the assertion of Theorem 4 in the main text, i.e., for any $f \in \mathcal{D}^2$ and compact subset $K \subset U_f$, there exist $W_1, \ldots, W_r \in \mathrm{Aff}$ and $\tau_1, \ldots, \tau_r \in \mathcal{S}_{\mathrm{c}}^2$ and $b \in \mathbb{R}^d$ such that $f(x) = W_1 \circ \tau_1 \circ \cdots \circ W_r \circ \tau_r(x)$ for all $x \in K$. Given this decomposition, we combine the $L^p$-universality (in the case A) or the sup-universality (in the case B) of $\mathrm{INN}_{\mathcal{G}}$ for $\mathcal{S}_{\mathrm{c}}^2$ with Proposition 6 to obtain the assertion of Theorem 1. $\qquad\square$

## B.1   From $\mathcal{D}^2$ to $\mathrm{Diff}_{\mathrm{c}}^2$

In this section, we describe how the approximation of $\mathcal{D}^2$ is reduced to that of $\mathrm{Diff}_{\mathrm{c}}^2$ when we are only concerned with its approximation on a compact set.

**Lemma 6.** *Let $f \colon U \to \mathbb{R}^d$ be an element of $\mathcal{D}^2$, and let $K \subset U$ be a compact set. Then, there exists $h \in \mathrm{Diff}_{\mathrm{c}}^2$ and an affine transform $W \in \mathrm{Aff}$ such that*

$$W \circ h|_K = f|_K.$$

*Proof of Lemma 6.* We denote the injections of $U$ and $f(U)$ into $\mathbb{R}^d$ by $\iota_1 \colon U \hookrightarrow \mathbb{R}^d$ and $\iota_2 \colon f(U) \hookrightarrow \mathbb{R}^d$, respectively. Since $U$ is $C^2$-diffeomorphic to $\mathbb{R}^d$ and $f$ is $C^2$-diffeomorphic, $f(U)$ is also $C^2$-diffeomorphic to $\mathbb{R}^d$. By applying Theorem 3.3 in [44] to $\iota_1 \circ f^{-1}|_{f(U)} \colon f(U) \to \mathbb{R}^d$ and the injection $\iota_2$, we can obtain diffeomorphisms $F_1 \colon f(U) \to \mathbb{R}^d$ and $F_2 \colon f(U) \to \mathbb{R}^d$ such that $F_1|_{f(K)} = f^{-1}|_{f(K)}$ and $F_2|_{f(K)} = \mathrm{Id}_{f(K)}$, where $\mathrm{Id}_{f(K)}$ denotes the identity map on $f(K)$. Let $F := F_2 \circ F_1^{-1} \colon \mathbb{R}^d \to \mathbb{R}^d$. By definition, we have $F|_K = f|_K$.

Take a sufficiently large open ball $B$ centered at 0 such that $K \subset B$. Let $W \in \mathrm{Aff}$ such that $W(x) = DF^{-1}(0)(x - F(0))$. Then by Lemma 7 below, we conclude that there exists a compactly supported diffeomorphism $h \colon \mathbb{R}^d \to \mathbb{R}^d$ such that $W \circ h|_K = F|_K = f|_K$. $\qquad\square$

**Lemma 7.** *Let $B_r \subset \mathbb{R}^d$ be an open ball of radius $r$ with origin 0, and let $f \colon B_r \to f(B_r) \subset \mathbb{R}^d$ be a $C^2$-diffeomorphism onto its image such that $f(0) = 0$ and $Df(0) = I$. Let $\varepsilon \in (0, r/2)$. Then there exists $h \in \mathrm{Diff}_{\mathrm{c}}^2$ such that $f(x) = h(x)$ for any $x \in B_{r-\varepsilon}$.*

*Proof.* Put $\delta := \varepsilon/(2r - \varepsilon)$, and define $I_\delta := (-1 - \delta, 1 + \delta)$. We define $F \colon B_{r-\varepsilon/2} \times I_\delta \to \mathbb{R}^d$ by

$$F(x,t) := \begin{cases} \frac{f(tx)}{t} & \text{if } t \neq 0, \\ x & \text{if } t = 0. \end{cases}$$

Let $U := F(B_{r-\varepsilon/2})$ and let $F^\dagger \colon U \times I_\delta \to B_{r-\varepsilon/2}$ such that $F^\dagger(F(x,t)) = x$ for any $(x,t) \in U$. Fix a compactly supported function on $\mathbb{R}^d \times I_\delta$ such that for $(x,t) \in F(\overline{B_{r-\varepsilon}} \times [-1,1])$, $\phi(x,t) = 1$, and for $(x,t) \notin U$ $\phi = 0$. Then we define $H \colon \mathbb{R}^d \times I_\delta \to \mathbb{R}^d$ by

$$H(x,t) := \phi(x,t) \frac{\partial F}{\partial t}(F^\dagger(x,t), t).$$

Since $f$ is $C^2$ diffeomorphism, there exists $L > 0$ such that for any $t \in I_\delta$, $\|H(x,t) - H(y,t)\| < L\|x - y\|$ with $x, y \in \mathbb{R}^d$. Thus the differential equation

$$\frac{dz}{dt} = H(z,t), \quad z(0) = x$$

has a unique solution $\phi_x(t)$. Then $h(x) := \phi_x(1)$ is the desired extension. $\qquad\square$

Here, we remark that Lemma 7 is a modified version of Lemma D.1 in Bernard et al. [44], with a correction to make it explicit that the extended diffeomorphism is compactly supported. Their Lemma D.1 does not explicitly state that it is compactly supported, but by Theorem 1.4 in Section 8 of Hirsch [45], it can be shown that the diffeomorphism is actually compactly supported.

## B.2   From $\mathrm{Diff}_\mathrm{c}^2$ to $\mathcal{S}_\mathrm{c}^\infty$ and permutations

The goal of this section is to show Theorem 5, which reduces the approximation problem of $\mathrm{Diff}_\mathrm{c}^2$ to that of $\mathcal{S}_\mathrm{c}^2$, and Lemma 11, which reduces from $\mathcal{S}_\mathrm{c}^2$ to $\mathcal{S}_\mathrm{c}^\infty$.

**Theorem 5.** *Let $f \in \mathrm{Diff}_\mathrm{c}^2$. Then there exist $\tau_1, \ldots, \tau_n \in \mathcal{S}_\mathrm{c}^2 \cap \mathrm{Diff}_\mathrm{c}^2$, and permutations of variables $\sigma_1, \ldots, \sigma_n \in \mathfrak{S}_d$, such that*
$$f = \tau_1 \circ \sigma_1 \circ \cdots \circ \tau_n \circ \sigma_n.$$

*Proof.* Combining Corollary 1, Lemma 8, and Lemma 9, we have the assertion.  □

We defer the statement and proof of Corollary 1, which describes the key properties of $\mathrm{Diff}_\mathrm{c}^2$, to Section C. In the remainder of this section, we describe Lemma 8, Lemma 9, and Lemma 11. First, Lemma 8 claims that the nearly-Id elements necessarily satisfy the condition of Lemma 9 below.

**Lemma 8.** *Let $A = (a_{i,j})_{i,j=1,\ldots,d}$ be a matrix. If $\|A - I_d\|_\mathrm{op} < 1$, then for $k = 1, \ldots, d$, the $k$-th trailing principal submatrix $A_k := (a_{i+k-1,j+k-1})_{i,j=1,\ldots,d-(k-1)}$ of $A$ is invertible. Here $I_d$ is a unit matrix of degree $d$.*

*Proof.* Let $v \in \mathbb{R}^{d-k+1}$ with $\|v\| = 1$, and put $w := (0, \ldots, 0, v) \in \mathbb{R}^d$. Then we have $1 > \|(A - I_d)w\|^2 \geq \|(A_k - I_k)v\|^2$. Thus $\|A_k - I_k\| < 1$. Since $\sum_{r=0}^\infty (I_k - A_k)^r$ absolutely converges, and it is identical to the inverse of $A_k$, we have that $A_k$ is invertible.  □

We apply the following lemma together with Lemma 8 to decompose nearly-Id elements into $\mathcal{S}_\mathrm{c}^2$ and permutations. For $a \in \mathbb{N}$, we denote the set of $a$-by-$a$ real-valued matrices by $M(a, \mathbb{R})$.

**Lemma 9.** *Let $r$ be a positive integer and $f \colon \mathbb{R}^d \to \mathbb{R}^d$ a compactly supported $C^r$-diffeomorphism. We write $f = (f_1, \ldots, f_d)$ with $f_i \colon \mathbb{R}^d \to \mathbb{R}$. For $k \in [d]$, let $\Delta_k^f(\boldsymbol{x}) \in M(d - (k-1), \mathbb{R})$ be the $k$-th trailing principal submatrix of Jacobian matrix of $f$, whose $(i, j)$ component is given by $\left( \frac{\partial f_{i+k-1}}{\partial x_{j+k-1}}(\boldsymbol{x}) \right)$ $(i, j = 1, \cdots, d - (k-1))$. We assume*
$$\det \Delta_k^f(x) \neq 0 \text{ for any } k \in [d] \text{ and } x \in \mathbb{R}^d.$$

*Then there exist compactly supported $C^r$-diffeomorphisms $F_1, \ldots, F_d \colon \mathbb{R}^d \to \mathbb{R}^d$ in the forms of*
$$F_i(\boldsymbol{x}) := (x_1, \ldots, x_{i-1}, h_i(\boldsymbol{x}), x_{i+1}, \ldots, x_d)$$

*for some $h_i \colon \mathbb{R}^d \to \mathbb{R}$ such that the identity holds:*
$$f = F_1 \circ \cdots \circ F_d.$$

*Proof.* The proof is based on induction. Suppose that $f$ is in the form of $f(\boldsymbol{x}) = (f_1(\boldsymbol{x}), \ldots, f_m(\boldsymbol{x}), x_{m+1}, \ldots, x_d)$. By means of induction with respect to $m$, we prove that there exist compactly supported $C^r$-diffeomorphisms $F_1, \ldots, F_m \colon \mathbb{R}^d \to \mathbb{R}^d$ in the forms of $F_i(\boldsymbol{x}) := (x_1, \ldots, x_{i-1}, h_i(\boldsymbol{x}), x_{i+1}, \ldots, x_d)$ for some $h_i \colon \mathbb{R}^d \to \mathbb{R}$ such that $f = F_1 \circ \cdots \circ F_m$.

In the case of $m = 1$, the above is clear. Assume that the statement is true in the case of any $k < m$. Define
$$F(x_1, \ldots, x_d) := (x_1, \ldots, x_{m-1}, f_m(\boldsymbol{x}), x_{m+1}, \ldots, x_d),$$
$$\tilde{f} := f \circ F^{-1}.$$

Note that $F$ is a compactly supported $C^r$-diffeomorphism from $\mathbb{R}^d$ to $\mathbb{R}^d$. In fact, compactly supportedness and surjectivity of $F$ comes from the compactly supportedness of $f$. More-over, since we have $\det DF_x = \frac{\partial f_m}{\partial x_m}(x) \neq 0$ for any $x \in \mathbb{R}^d$ by the assumption on $f$, $F$

is injective and is a $C^r$-diffeomorphism from $\mathbb{R}^d$ to $\mathbb{R}^d$ by inverse function theorem. Therefore, $\tilde{f}$ is also a $C^r$-diffeomorphism from $\mathbb{R}^d$ to $\mathbb{R}^d$. We show that $\tilde{f}$ is of the form $\tilde{f}(\boldsymbol{x}) = (g_1(\boldsymbol{x}), \cdots, g_{m-1}(\boldsymbol{x}), x_m, \cdots, x_d)$ for some $C^r$-functions $g_i \colon \mathbb{R}^d \to \mathbb{R}$ $(i = 1, \cdots, m-1)$ satisfying $\det \Delta_k^{\tilde{f}}(x) \neq 0$ for any $x \in \mathbb{R}^d$ and $k \in [d]$. From Lemma 10, there exist $g_i, h \in C^r(\mathbb{R}^d)$ $(i = 1, \cdots, m)$ such that

$$f^{-1}(\boldsymbol{x}) = (g_1(\boldsymbol{x}), \cdots, g_m(\boldsymbol{x}), x_{m+1}, \cdots, x_d)$$
$$F^{-1}(\boldsymbol{x}) = (x_1, \cdots, x_{m-1}, h(\boldsymbol{x}), x_{m+1}, \cdots, x_d).$$

Then we have

$$\tilde{f}^{-1}(\boldsymbol{x}) = F \circ f^{-1}(\boldsymbol{x}) = (g_1(\boldsymbol{x}), \cdots, g_{m-1}(\boldsymbol{x}), f_m(f^{-1}(\boldsymbol{x})), x_{m+1}, \cdots, x_d)$$
$$= (g_1(\boldsymbol{x}), \cdots, g_{m-1}(\boldsymbol{x}), x_m, \cdots, x_d).$$

Therefore, from Lemma 10, $\tilde{f}$ is of the following form

$$\tilde{f}(x) = f \circ F^{-1}(x) = (f_1 \circ F^{-1}(x), \cdots, f_{m-1} \circ F^{-1}(x), x_m, \cdots, x_d).$$

Moreover, by the form of $F^{-1}$ and $f$, we have $D\tilde{f}(x) = Df(F^{-1}(x)) \circ DF^{-1}(x)$ and

$$Df = \begin{pmatrix} A & \\ & I \end{pmatrix}, \quad D(F^{-1}) = \begin{pmatrix} I_{m-1} & & \\ \frac{\partial h}{\partial x_1} & \cdots & \frac{\partial h}{\partial x_d} \\ & & I_{d-m} \end{pmatrix}$$

for some $A \in M(m, \mathbb{R})$ with all the trailing principal minors nonzero. Therefore, we obtain $\det \Delta_k^f(x) \neq 0$ for any $x \in \mathbb{R}^d$ and $k \in [d]$. Here, by the assumption of the induction, there exist compactly supported $C^r$-diffeomorphisms $F_i \colon \mathbb{R}^d \to \mathbb{R}^d$ and $h_i \in C^r(\mathbb{R}^d)$ $(i = 1, \cdots, m-1)$ such that

$$\tilde{f} = F_1 \circ \cdots \circ F_{m-1}, \quad F_i(\boldsymbol{x}) = (x_1, \cdots x_{i-1}, h_i(x), x_{i+1}, \cdots, x_d).$$

Thus $f = \tilde{f} \circ F$ has a desired form. $\qquad\square$

**Lemma 10.** *Let $r$ be a positive integer and $f \colon \mathbb{R}^d \to \mathbb{R}^d$ $C^r$-diffeomorphism of the form*

$$f(\boldsymbol{x}) := (f_1(\boldsymbol{x}), \cdots, f_m(\boldsymbol{x}), x_{m+1}, \cdots, x_d),$$

*where $f_i \colon \mathbb{R}^d \to \mathbb{R}$ belongs to $C^r(\mathbb{R}^d)$ $(i = 1, \cdots, m)$. Then the inverse map $f^{-1}$ becomes of the form*

$$f^{-1}(\boldsymbol{x}) = (g_1(\boldsymbol{x}), \cdots, g_m(\boldsymbol{x}), x_{m+1}, \cdots x_d),$$

*where $g_i \colon \mathbb{R}^d \to \mathbb{R}$ belongs to $C^r(\mathbb{R}^d)$ for $i = 1, \cdots, m$.*

*Proof.* We write $f^{-1}(\boldsymbol{x}) = (h_1(\boldsymbol{x}), \cdots, h_d(\boldsymbol{x}))$, where $h_i \in C^r(\mathbb{R}^d)$ $(i = 1, \cdots, d)$. Then by the definition of the inverse map, the identity

$$(x_1, \cdots, x_d) = f \circ f^{-1}(\boldsymbol{x}) = (f_1(h_1(\boldsymbol{x})), \cdots, f_m(h_m(\boldsymbol{x})), h_{m+1}(\boldsymbol{x}), \cdots, h_d(\boldsymbol{x}))$$

holds for any $\boldsymbol{x} \in \mathbb{R}^d$, which implies that we obtain $h_i(x) = x_i$ $(i = m+1, \cdots, d)$. This completes the proof of the lemma. $\qquad\square$

The following Lemma 11 is used in the main text in reducing the approximation problem from $\mathcal{S}_c^2$ to $\mathcal{S}_c^\infty$. We say that $f \colon \mathbb{R}^d \to \mathbb{R}$ is a *locally $L^p$-function* if $\int_K |f(x)|^p dx < \infty$ holds for any compact set $K \subset \mathbb{R}^d$.

**Definition 7** (Last-increasing)**.** We say that a map $f \colon \mathbb{R}^d \to \mathbb{R}$ is *last-increasing* if, for any $(a_1, \ldots, a_{d-1}) \in \mathbb{R}^{d-1}$, the function $f(a_1, \ldots, a_{d-1}, x)$ is strictly increasing with respect to $x$.

**Lemma 11.** *Let $\tau \colon \mathbb{R}^d \to \mathbb{R}$ be a last-increasing locally $L^p$-function. Then for any compact subset $K \subset \mathbb{R}^d$ and any $\varepsilon > 0$, there exists a last-increasing $C^\infty$-function $\tilde{\tau} \colon \mathbb{R}^d \to \mathbb{R}$ satisfying*

$$\|\tau - \tilde{\tau}\|_{p,K} < \varepsilon.$$

*Moreover, if $\tau$ is continuous, there exists a last-increasing $C^\infty$-function $\tilde{\tau}$ such that*

$$\|\tau - \tilde{\tau}\|_{\mathrm{sup}, K} < \varepsilon.$$

*Proof.* Let $\phi : \mathbb{R}^d \to \mathbb{R}$ be a compactly supported non-negative $C^\infty$-function with $\int |\phi(x)| dx = 1$ such that for any $(a_1, \ldots, a_{d-1}) \in \mathbb{R}^{d-1}$, the function $\phi(a_1, \ldots, a_{d-1}, x)$ of $x$ is even and decreasing on $\{x > 0 : \phi(a_1, \ldots, a_{d-1}, x) > 0\}$. For $t > 0$, we define $\phi_t(x) := t^{-d}\phi(x/t)$. Then we see that $\tau_t := \phi_t * \tau$ is a $C^\infty$-function. We take any $\boldsymbol{a} \in \mathbb{R}^{d-1}$. We verify that $\tau_t(\boldsymbol{a}, x_d)$ is strictly increasing with respect to $x_d$. Take any $x_d, x_d' \in \mathbb{R}$ satisfying $x_d > x_d'$. Since $\tau$ is strictly increasing, we have

$$\tau_t(\boldsymbol{a}, x_d) - \tau_t(\boldsymbol{a}, x_d') = \int_{\mathbb{R}^d} \phi_t(x)(\tau((\boldsymbol{a}, x_d) - x) - \tau((\boldsymbol{a}, x_d') - x)) dx > 0.$$

Thus for any $(a_1, \ldots, a_{d-1}) \in \mathbb{R}^{d-1}$, the $C^\infty$-function $\tau_t(a_1, \ldots, a_{d-1}, x)$ is strictly increasing for with respect to $x$.

Next, take any compact subset $K \subset \mathbb{R}^d$. We show $\|\tau_t - \tau\|_{p,K} \to 0$ as $t \to 0$. We prove $\tau_t$ converges $\tau$ as $t \to 0$. Take $R > 0$ satisfying $K \subset B(R) := \{x \in \mathbb{R}^d : |x| \le R\}$. We assume $0 < t < 1$. Then we have $\phi_t * \tau = \phi_t * (\mathbf{1}_{B(R+1)}\tau)$. Since we have $\mathbf{1}_{B(R+1)}\tau \in L^p(\mathbb{R}^d)$, we obtain

$$\begin{aligned}
\|\phi_t * \tau - \tau\|_{p,K} &= \|\phi_t * (\mathbf{1}_{B(R+1)}\tau) - \mathbf{1}_{B(R+1)}\tau\|_{p,K} \\
&\le \|\phi_t * (\mathbf{1}_{B(R+1)}\tau) - \mathbf{1}_{B(R+1)}\tau\|_{p,\mathbb{R}^d} \to 0 \quad (t \to 0).
\end{aligned}$$

Here, we used a property of mollifier $\phi_t$ (see Theorem 8.14 in [46] for example).

Next, we consider the sup-approximation when $\tau$ is continuous. By direct computation, we have

$$\begin{aligned}
\sup_{y \in K} |\tau_t(y) - \tau(y)| &\le \sup_{y \in K} \int_{\mathbb{R}^d} |\phi(x)| \cdot |\tau(y - tx) - \tau(y)| dx \\
&\le C \sup_{(x,y) \in \mathrm{supp}(\phi) \times K} |\tau(y - tx) - \tau(y)| \to 0 \quad (t \to 0).
\end{aligned}$$

Here $C := \sup_{x \in \mathbb{R}^d} |\phi(x)|$. Thus in both cases above, By taking sufficiently small $t$, we obtain the desired $C^\infty$-function $\tilde{\tau} = \tau_t$. $\qquad\square$

# C   Key properties of diffeomorphisms on $\mathbb{R}^d$: From $\mathrm{Diff}_c^2$ to Nearly-$\mathrm{Id}$

This section explains the reduction of the universality for $\mathrm{Diff}_c^2$ to Nearly-$\mathrm{Id}$ elements. The reduction involves a structure theorem from the field of differential geometry. The results of this section are used as a building block for the proofs in Section B.2.

**Definition 8** (Compactly supported diffeomorphism). The diffeomorphism $f$ on $\mathbb{R}^d$ is *compactly supported* if there exists a compact subset $K \subset \mathbb{R}^d$ such that for any $x \notin K$, $f(x) = x$. We denote by $\mathrm{Diff}_c^2$ the space of compactly supported $C^2$-diffeomorpshisms.

The set $\mathrm{Diff}_c^2$ constitutes a group whose group operation is the function composition. Moreover, $\mathrm{Diff}_c^2$ is a topological group with respect to the *Whitney topology* [27, Proposition 1.7.(9)]. Then there is a crucial structure theorem of $\mathrm{Diff}_c^2$ attributed to Herman, Thurston [28], Epstein [29], and Mather [30, 31]:

**Fact 2.** *The group $\mathrm{Diff}_c^2$ is simple, i.e., any normal subgroup $H \subset \mathrm{Diff}_c^2$ is either $\{\mathrm{Id}\}$ or $\mathrm{Diff}_c^2$.*

The assertion is proven in Mather [31] for the connected component containing $\mathrm{Id}$, instead of the entire set of compactly-supported $C^2$-diffeomorphisms when the domain space is a general manifold instead of $\mathbb{R}^d$. In the special case of $\mathbb{R}^d$, the connected component containing $\mathrm{Id}$ is shown to be $\mathrm{Diff}_c^2$ itself [27, Example 1.15], hence Fact 2 follows. For details, see [27, Corollary 3.5 and Example 1.15].

As a side note, the assertion of Theorem 2 is proved to hold generally for $C^r$-diffeomorphisms only except for $r = d + 1$ [27]. Nevertheless, this exception does not cause any problem in our proof, because we apply it with $r = 2$ and $d \ge 2$. The limitation only means that the structure of $C^2$-diffeomorphisms is better understood than that of $C^{d+1}$-diffeomorphisms. Also note that this exception does not affect the approximation capability for $C^{d+1}$-diffeomorphisms either as they are contained in $C^2$ where we perform our theoretical analyses. For the details of mathematical ingredients, see [47].

Here, we provide a precise definition of the *flow endpoints* introduced in Section 4.1.

**Definition 9** (Flow endpoints). A *flow endpoint* is an element of $\mathrm{Diff}_c^2$ which can be represented as $\phi(1)$, where $\phi : [0,1] \to \mathrm{Diff}_c^2$ is a continuous map such that $\phi(0) = \mathrm{Id}$ and that $\phi$ is additive, namely, $\phi(s) \circ \phi(t) = \phi(s+t)$ for any $s, t \in [0,1]$ with $s + t \in [0,1]$.

We use Fact 2 to prove that a compactly supported diffeomorphism can be represented as a composition of flow endpoints in $\mathrm{Diff}_c^2$. The following lemma is a restatement of Lemma 2 in the main text.

**Lemma 12.** *Let $S \subset \mathrm{Diff}_c^2$ be the set of all flow endpoints. Then, $\mathrm{Diff}_c^2$ coincides with the set of finite compositions of elements in $S$ defined by*

$$H := \{g_1 \circ \cdots \circ g_n : n \geq 1, g_1, \ldots, g_n \in S\}.$$

*Proof.* In view of Fact 2, it is enough to show that $H$ forms a subgroup, that it is normal, and that it is non-trivial.

First, we prove the $H$ consists a subgroup of $\mathrm{Diff}_c^2$. By definition, for any $g, h \in H$, it is immediate to show that $g \circ h \in H$. We prove that $H$ is closed under inversion. For this, it suffices to show that $S$ is closed under inversion. Let $g = \phi(1) \in S$. Consider the map $\varphi : [0,1] \to \mathrm{Diff}_c^2$ defined by $\varphi(t) := (\phi(t))^{-1}$. Since $\mathrm{Diff}_c^2$ is a topological group [27, Proposition 1.7.(9)], $\varphi$ is continuous. Moreover, it is immediate to show that $\varphi$ is additive in the sense of Definition 9, and that $\varphi(0) = \mathrm{Id}$. Thus, $g^{-1} = \varphi(1)$ is an element of $S$.

Next, we prove $H$ is normal. It suffice to show that $S$ is closed under conjugation since the conjugation $g \mapsto hgh^{-1}$ is a group homomorphism on $\mathrm{Diff}_c^2$. Let $g = \phi(1) \in S$, where $\phi : [0,1] \to \mathrm{Diff}_c^2$ is a continuous map associated to $g$. Then, we define a $\Phi : \mathbb{R}^d \times [0,1] \to \mathbb{R}^d$ by $\Phi(x,t) = \phi(t)(x)$. We call $\Phi$ a flow associated with $g$. We take arbitrary $h \in \mathrm{Diff}_c^2$. Then, the function $\Phi' : \mathbb{R}^d \times [0,1]$ defined by $\Phi'(\cdot, s) := h^{-1} \circ \Phi(\cdot, s) \circ h$ is a flow associated with $h^{-1}gh$, which means $h^{-1}gh \in S$, i.e., $S$ is closed under conjugation.

Finally, we show $H$ is nontrivial. It suffice to show that $S$ includes a non-identity element. Let $\psi : \mathbb{R} \to \mathrm{O}(d)$ be a nontrivial homomorphism of Lie groups, where $\mathrm{O}(d)$ is a orthogonal group of degree $d$. Such $\psi$ exists, for example, let $\psi(t) := \exp(tA)$ for some nonzero skew-symmetric matrix $A$, namely, $A^\top = -A$. Let $u : [0, \infty) \to \mathbb{R}$ be a compactly supported $C^\infty$ function such that its support does not include 0. Then, We define $\Phi : \mathbb{R}^d \times [0,1] \to \mathbb{R}^d$ by $\Phi(x,t) := \psi(u(|x|)t)x$. Then, $\Phi$ is the flow associated with $\Phi(\cdot, 1) \in S$, that is a non-identity element.

$\square$

**Definition 10** (Nearly-Id elements). Let $f \in \mathrm{Diff}_c^2$. We say $f$ is *nearly*-Id if, for any $x \in \mathbb{R}^d$, the Jacobian $Df$ of $f$ at $x$ satisfies

$$\|Df(x) - I\|_{\mathrm{op}} < 1,$$

where $I$ is the unit matrix.

**Corollary 1.** *For any $f \in \mathrm{Diff}_c^2$, there exist finite elements $g_1, \ldots, g_r \in \mathrm{Diff}_c^2$ such that $f = g_r \circ \cdots \circ g_1$ and $g_i$ is nearly-Id for any $i \in [r]$.*

*Proof.* Let $S$ be the subset of $\mathrm{Diff}_c^2$ as defined above. Therefore, by Lemma 12, there exist $h_1, \ldots, h_m \in S$ such that $f = h_m \circ \cdots \circ h_1$. For $i \in [m]$, let $\phi_i$ be a flow associated with $h_i$. Since $[0,1] \ni t \mapsto \Phi_i(\cdot, t) \in \mathrm{Diff}_c^2$ is continuous with respect to Whitney topology and $\Phi_i(\cdot, 0)$ is the identity function, we can take a sufficiently large $n$ such that $\tilde{h}_i := \Phi_i(\cdot, 1/n)$ is nearly-Id. By the additive property of $\Phi_i$, we have

$$f = h_m \circ \cdots \circ h_1 = \underbrace{\tilde{h}_m \circ \cdots \circ \tilde{h}_m}_{n \text{ times}} \circ \cdots \circ \underbrace{\tilde{h}_1 \circ \cdots \circ \tilde{h}_1}_{n \text{ times}},$$

which completes the proof of the corollary.

$\square$

# D  Proof of Theorem 2: $L^p$-universality of $\mathrm{INN}_{\mathcal{H}\text{-ACF}}$

In this section, we provide the proof details of Theorem 2 in the main text. The correspondence between this section and Section 4.2 in the main text is as follows: Steps 1, 2, 3 correspond to Section D.1, Step 4 corresponds to Section D.2, and Step 5 is justified by Proposition 6 in Section F.

## D.1 Approximation of general elements of $\mathcal{S}_c^0$

In this section, we prove the following lemma to construct an approximator for an arbitrary element of $\mathcal{S}_c^0$ (hence for $\mathcal{S}_c^\infty$) within $\mathrm{INN}_{\mathcal{H}\text{-}\mathrm{ACF}}$. It is based on Lemma 14 proved in Section D.2, which corresponds to a special case.

Here, we rephrase Theorem 2 as in the following:

**Lemma 13** ($L^p$-universality of $\mathrm{INN}_{\mathcal{H}\text{-}\mathrm{ACF}}$ for compactly supported $\mathcal{S}_c^\infty$). *Let $p \in [1, \infty)$. Assume $\mathcal{H}$ is a sup-universal approximator for $C_c^\infty(\mathbb{R}^{d-1})$ and that it consists of piecewise $C^1$-functions. Let $f \in \mathcal{S}_c^0$, $\varepsilon > 0$, and $K \subset \mathbb{R}^d$ be a compact subset. Then, there exists $g \in \mathrm{INN}_{\mathcal{H}\text{-}\mathrm{ACF}}$ such that $\|f - g\|_{p,K} < \varepsilon$.*

*Proof.* Since we can take $a > 0$, $b \in \mathbb{R}$ satisfying $aK + b \subset [0,1]^d$, it is enough to prove the assertion for the case $K = [0,1]^d$.

Next, we show that we can assume that for any $(\boldsymbol{x}, y) \in \mathbb{R}^d$, $u(\boldsymbol{x}, 0) = 0$ and $u(\boldsymbol{x}, 1) = 1$ for any $\boldsymbol{x} \in \mathbb{R}^{d-1}$. Since $u(\boldsymbol{x}, \cdot)$ is a diffeomorphism, we have $u(\boldsymbol{x}, 0) \neq u(\boldsymbol{x}, 1)$ for any $x \in \mathbb{R}$. By the continuity of $f$, either of $u(\boldsymbol{x}, 0) > u(\boldsymbol{x}, 1)$ for all $\boldsymbol{x} \in [0,1]^{d-1}$ or $u(\boldsymbol{x}, 0) < u(\boldsymbol{x}, 1)$ for all $x \in [0,1]^{d-1}$ holds. Without loss of generality, we assume the latter case holds (if the former one holds, we just switch $u(\boldsymbol{x}, 0)$ and $u(\boldsymbol{x}, 1)$). We define $s(\boldsymbol{x}) = -\log(u(\boldsymbol{x}, 1) - u(\boldsymbol{x}, 0))$ and $t(\boldsymbol{x}) = -u(\boldsymbol{x}, 0)(u(\boldsymbol{x}, 1) - u(\boldsymbol{x}, 0))^{-1}$. By a direct computation, we have

$$\Psi_{d-1,s,t} \circ f(\boldsymbol{x}, y) = \left( \boldsymbol{x}, \frac{u(\boldsymbol{x}, y) - u(\boldsymbol{x}, 0)}{u(\boldsymbol{x}, 1) - u(\boldsymbol{x}, 0)} \right) =: (\boldsymbol{x}, u_0(\boldsymbol{x}, y)).$$

In particular, $\Psi_{s,t} \circ f(\boldsymbol{x}, 0) = (\boldsymbol{x}, 0)$ and $\Psi_{s,t} \circ s(\boldsymbol{x}, 1) = (\boldsymbol{x}, 1)$ hold. , and the map $y \mapsto u_0(\boldsymbol{x}, y)$ is a diffeomorphism for each $\boldsymbol{x}$. Thus if we prove the existence of an approximator for $\Psi_{s,t} \circ f$, by Proposition 6, we can arbitrarily approximate $f$ itself.

For $\underline{k} := (k_1, \ldots, k_{d-1}) \in \mathbb{Z}^{d-1}$ and $n \in \mathbb{N}$, we define $(\underline{k})_n := \sum_{i=1}^d k_i n^{i-1} \in \{0, \ldots, n^d - 1\}$, that is, $\underline{k}$ is the $n$-adic expansion of $(\underline{k})_n$. For any $n \in \mathbb{N}$, define the following discontinuous ACF: $\psi_n \colon [0,1]^d \to [0,1]^{d-1} \times [0, n^d]$ by

$$\psi_n(\boldsymbol{x}, y) := \left( \boldsymbol{x}, y + \sum_{k_1, \cdots, k_{d-1}=0}^{n-1} (\underline{k})_n 1_{\Delta_{\underline{k}+1}^n}(\boldsymbol{x}) \right),$$

where $\underline{k} := (k_1, \ldots, k_d)$ and $\underline{k} + 1 := (k_1 + 1, \ldots, k_d + 1)$. We take an increasing function $v_n \colon \mathbb{R} \to \mathbb{R}$ that is smooth outside finite points such that

$$v_n(z) := \begin{cases} u\left( \frac{k_1}{n}, \cdots, \frac{k_{d-1}}{n}, z - (\underline{k})_n \right) + (\underline{k})_n & \text{if } z \in [(\underline{k})_n, (\underline{k})_n + 1) \\ z & \text{if } z \notin [0, n^d). \end{cases}$$

We consider maps $h_n$ on $[0,1]^{d-1} \times [0, n^d]$ and $f_n \colon [0,1]^d \to [0,1]^d$ defined by

$$h_n(\boldsymbol{x}, z) := (\boldsymbol{x}, v_n(z)),$$
$$f_n := \psi_n^{-1} \circ h_n \circ \psi_n.$$

Then we have the following claim.
**Claim.** For all $k_1, \cdots, k_{d-1} = 0, \cdots, n-1$, we have

$$f_n(\boldsymbol{x}, y) = \left( \boldsymbol{x}, u\left( \frac{k_1}{n}, \ldots, \frac{k_{d-1}}{n}, y \right) \right)$$

on $\prod_{i=1}^{d-1} [\frac{k_i}{n}, \frac{k_i+1}{n}) \times [0, 1)$.

In fact, we have

$$\begin{aligned}
f_n(\boldsymbol{x}, y) &= \psi_n^{-1} \circ h_n \circ \psi_n(\boldsymbol{x}, y) \\
&= \psi_n^{-1} \circ h_n(\boldsymbol{x}, y + (\underline{k})_n) \\
&= \psi_n^{-1}(\boldsymbol{x}, v_n(y + (\underline{k})_n)) \\
&= \psi_n^{-1}\left(\boldsymbol{x}, u\left(\frac{k_1}{n}, \ldots, \frac{k_{d-1}}{n}, y\right) + (\underline{k})_n\right) \\
&= \left(\boldsymbol{x}, u\left(\frac{k_1}{n}, \ldots, \frac{k_{d-1}}{n}, y\right)\right).
\end{aligned}$$

Therefore, the claim above has been proved. Hence we see that $\|f - f_n\|_{\sup, K} \to 0$ as $n \to \infty$. By Lemma 14 below and the universal approximation property of $\mathcal{H}$, for any compact subset $K$ and $\varepsilon > 0$, there exist $g_1, g_2, g_3 \in \text{INN}_{\mathcal{H}\text{-ACF}}$ such that $\|g_1 - \psi_n^{-1}\|_{p,K} < \varepsilon$, $\|g_2 - h_n\|_{p,K} < \varepsilon$, and $\|g_3 - \psi_n\|_{p,K} < \varepsilon$. Thus by Proposition 6, for any compact $K$ and $\varepsilon > 0$, there exists $g \in \text{INN}_{\mathcal{H}\text{-ACF}}$ such that $\|g - f\|_{p,K} < \varepsilon$. $\qquad\square$

## D.2 Special case: Approximation of coordinate-wise independent transformation

In this section, we show the lemma claiming that special cases of single-coordinate transformations, namely coordinate-wise independent transformations, can be approximated by the elements of $\text{INN}_{\mathcal{H}\text{-ACF}}$ given sufficient representational power of $\mathcal{H}$.

**Lemma 14.** *Let $p \in [1, \infty)$. Assume $\mathcal{H}$ is a sup-universal approximator for $C_c^\infty(\mathbb{R}^{d-1})$ and that it consists of piecewise $C^1$-functions. Let $u : \mathbb{R} \to \mathbb{R}$ be a continuous increasing function. Let $f : \mathbb{R}^d \to \mathbb{R}^d; (\boldsymbol{x}, y) \mapsto (\boldsymbol{x}, u(y))$ where $\boldsymbol{x} \in \mathbb{R}^{d-1}$ and $y \in \mathbb{R}$. For any compact subset $K \subset \mathbb{R}^d$ and $\varepsilon > 0$, there exists $g \in \text{INN}_{\mathcal{H}\text{-ACF}}$ such that $\|f - g\|_{p,K} < \varepsilon$.*

*Proof.* We may assume without loss of generality, in light of Lemma 11, that $u$ is a $C^\infty$-diffeomorphism on $\mathbb{R}$ and that the inequality $u'(y) > 0$ holds for any $y \in \mathbb{R}$. Furthermore, we may assume that $u$ is compactly supported (i.e., $u(y) = y$ outside a compact subset of $\mathbb{R}$) without loss of generality because we can take a compactly supported diffeomorphism $\tilde{u}$ and $a, b \in \mathbb{R}$ ($a \neq 0$) such that $a\tilde{u} + b = u$ on any compact set containing $K$ by Lemma 6, and the scaling $a$ and the offset $b$ can be realized by the elements of $\text{INN}_{\mathcal{H}\text{-ACF}}$.

Fix $\delta \in (0, 1)$. We define the following functions:

$$\begin{aligned}
\psi_0(\boldsymbol{x}, y) &:= (\boldsymbol{x}_{\leq d-2}, u'(y)x_{d-1}, y) \\
&= (\boldsymbol{x}_{\leq d-2}, \exp(\log u'(y))x_{d-1}, y), \\
\psi_1(\boldsymbol{x}, y) &:= \left(\boldsymbol{x}_{\leq d-2}, x_{d-1} + \delta^{-1}(u(y) - y), y\right), \\
\psi_2(\boldsymbol{x}, y) &:= (\boldsymbol{x}_{\leq d-2}, x_{d-1}, y + \delta x_{d-1}), \\
\psi_3(\boldsymbol{x}, y) &:= \left(\boldsymbol{x}_{\leq d-2}, x_{d-1} - \delta^{-1}(y - u^{-1}(y)), y\right),
\end{aligned}$$

where we denote $\boldsymbol{x} = (x_1, \ldots, x_{d-1}) \in \mathbb{R}^{d-1}$. First, we show that $\|f - \psi_3 \circ \psi_2 \circ \psi_1 \circ \psi_0\|_{\sup, K} \to 0$ as $\delta \to 0$. By a direct computation, we have

$$\begin{aligned}
\psi_3 \circ \psi_2 \circ \psi_1(\boldsymbol{x}, y) &= \psi_3 \circ \psi_2(\boldsymbol{x}_{\leq d-2}, x_{d-1} + \delta^{-1}(u(y) - y), y) \\
&= \psi_3(\boldsymbol{x}_{\leq d-2}, x_{d-1} + \delta^{-1}(u(y) - y), y + \delta(x_{d-1} + \delta^{-1}(u(y) - y))) \\
&= \psi_3(\boldsymbol{x}_{\leq d-2}, x_{d-1} + \delta^{-1}(u(y) - y), \delta x_{d-1} + u(y)) \\
&= (\boldsymbol{x}_{\leq d-2}, x_{d-1} - \delta^{-1}(\delta x_{d-1} + u(y) - u^{-1}(\delta x_{d-1} + u(y))), \delta x_{d-1} + u(y)) \\
&= (\boldsymbol{x}_{\leq d-2}, \delta^{-1}u^{-1}(\delta x_{d-2} + u(y)) - \delta^{-1}y, u(y) + \delta x_{d-1}),
\end{aligned}$$

where $\boldsymbol{x} = (x_1, \ldots, x_{d-1}) \in \mathbb{R}^{d-1}$. Since $u \in C^\infty([-r, r])$ where $r = \max_{(\boldsymbol{x}, y) \in K} |y|$, by applying Taylor's theorem, there exists a function $R(\boldsymbol{x}, y; \delta)$ and $C = C([-r, r], u) > 0$ such that

$$u^{-1}(u(y) + \delta x) = y + u'(y)^{-1}\delta x + R(\boldsymbol{x}, y; \delta)(\delta x)^2 \quad \text{and} \quad \sup_{\delta \in (0,1)} |R(\boldsymbol{x}, y; \delta)| \leq C$$

for all $(\boldsymbol{x}, y) \in K$. Therefore, we have

$$\psi_3 \circ \psi_2 \circ \psi_1 \circ \psi_0(\boldsymbol{x}, y) = (\boldsymbol{x}, u(y)) + \delta(R(\boldsymbol{x}, u'(y)x_{d-1}; \delta)\boldsymbol{x}_{\leq d-1}, u'(y)x_{d-1}).$$

For any compact subset $K$, the last term uniformly converges to 0 as $\delta \to 0$ on $K$.

Assume $\delta$ is taken to be small enough. Now, we approximate $\psi_3 \circ \cdots \circ \psi_0$ by the elements of $\text{INN}_{\mathcal{H}\text{-ACF}}$. Since $u$ is a compactly-supported $C^\infty$-diffeomorphism on $\mathbb{R}$, the functions $(\boldsymbol{x}_{\leq d-2}, y) \mapsto \log u'(y)$, $(\boldsymbol{x}_{\leq d-2}, y) \mapsto u(y) - y$, and $(\boldsymbol{x}_{\leq d-2}, y) \mapsto y - u^{-1}(y)$, each appearing in $\psi_0, \psi_1, \psi_3$, respectively, belong to $C_c^\infty(\mathbb{R}^{d-1})$. On the other hand, $\psi_2$ can be realized by $\text{GL} \subset \text{Aff}$. Therefore, combining the above with the fact that $\mathcal{H}$ is a sup-universal approximator for $C_c^\infty(\mathbb{R}^{d-1})$, we have that for any compact subset $K' \subset \mathbb{R}^d$ and any $\varepsilon > 0$, there exist $\phi_0, \ldots, \phi_3 \in \text{INN}_{\mathcal{H}\text{-ACF}}$ such that $\|\psi_i - \phi_i\|_{\sup, K'} < \varepsilon$. In particular, we can find $\phi_0, \ldots, \phi_3 \in \text{INN}_{\mathcal{H}\text{-ACF}}$ such that $\|\psi_i - \phi_i\|_{p, K'} < \varepsilon$.

Now, recall that $\mathcal{H}$ consists of piecewise $C^1$-functions as well as $\psi_i$ ($i = 0, \ldots, 3$). Moreover, $\psi_0, \psi_1, \psi_3$ are compactly supported while $\psi_2 \in \text{GL}$, hence they are Lipschitz continuous outside a bounded open subset. Therefore, by Proposition 6, we have the assertion of the lemma.

$\square$

# E   Locally bounded maps and piecewise diffeomorphisms

In this section, we provide the notions of locally bounded maps and piecewise $C^1$-maps. These notions are used to state the regularity conditions on the CF layers in Theorem 1 and to prove the results in Section F.

## E.1   Definition of locally bounded maps

Here, we provide the definition of locally bounded maps. It is a very mild condition that is satisfied in most cases of practical interest, e.g., by continuous maps.

**Definition 11** (Locally bounded maps)**.** Let $f$ be a map from $\mathbb{R}^m$ to $\mathbb{R}^n$. We say $f$ is *locally bounded* if for each point $\boldsymbol{x} \in \mathbb{R}^m$, there exists a neighborhood $U$ of $\boldsymbol{x}$ such that $f$ is bounded on $U$.

As a special case, continuous maps are locally bounded; take an open ball $U$ centered at $\boldsymbol{x}$ and take a compact set containing $U$ to see that $f$ is bounded on $U$.

## E.2   Definition and properties of piecewise $C^1$-mappings

In this section, we give the definition of piecewise $C^1$-mappings and their properties. Examples of piecewise $C^1$-diffeomorphisms appearing in the paper include the $\mathcal{H}$-ACF with $\mathcal{H}$ being MLPs with ReLU activation.

**Definition 12** (piecewise $C^1$-mappings)**.** Let $f : \mathbb{R}^m \to \mathbb{R}^n$ be a measurable map. We say $f$ is a *piecewise $C^1$-mapping* if there exists a mutually disjoint family of (at most countable) open subsets $\{V_i\}_{i \in I}$ such that

- $\text{vol}(\mathbb{R}^d \setminus U_f) = 0$,

- for any $i \in I$, there exists an open subset $W_i$ containing the closure $\overline{V_i}$ of $V_i$, and $C^1$-mapping $\tilde{f}_i : W_i \to \mathbb{R}^d$ such that $\tilde{f}_i|_{V_i} = f|_{V_i}$, and

- for any compact subset $K$, $\#\{i \in I : V_i \cap K \neq \emptyset\} < \infty$.

where we denote $U_f := \bigsqcup_{i \in I} V_i$, and $\#(\cdot)$ denotes the cardinality of a set.

We remark that piecewise $C^1$-mappings are essentially locally bounded in the sense that for any compact set $K \subset \mathbb{R}^d$, $\text{ess.sup}_K \|f\| = \|f\|_{\sup, K \cap U_f} < \infty$. Then we define a piecewise $C^1$-diffeomorphisms:

**Definition 13** (piecewise $C^1$-diffeomorphisms)**.** Let $f : \mathbb{R}^d \to \mathbb{R}^d$ be a piecewise $C^1$-mapping. We say $f$ is a *piecewise $C^1$-diffeomorphism* if

1. the image of nullset via $f$ is also a nullset,

2. $f|_{U_f}$ is injective, and for $i \in I$, $\tilde{f}_i$ is a $C^1$-diffeomorphism from $W_i$ onto $\tilde{f}_i(W_i)$,

3. $\mathrm{vol}\left(\mathbb{R}^d \setminus f(U_f)\right) = 0$, and

4. for any compact subset $K$, $\#\{i \in I : f(V_i) \cap K \neq \emptyset\} < \infty$.

We summarize the basic properties of piecewise $C^1$-diffeomorphisms in the proposition below:

**Proposition 4.** *Let $f$ and $g$ be piecewise $C^1$-diffeomorphisms. Then, we have the following.*

1. *There exists a piecewise $C^1$-diffeomorphism $f^\dagger$ such that $f(f^\dagger(x)) = x$ for $x \in U_{f^\dagger}$ and $f^\dagger(f(y)) = y$ for $y \in U_f$.*

2. *For any $h \in L^1$, we have $\int h(x)dx = \int h(f(x))|Df(x)|dx$, where $|Df(x)|$ is the absolute value of the determinat of the Jacobian matrix of $f$ at $x$.*

3. *For any compact subset $K$, $f^{-1}(K) \cap U_f$ is a bounded subset.*

4. *For any nullset $F$, then $f^{-1}(F)$ is also a nullset.*

5. *For any measurable set $E$ and any compact set $K$, $f^{-1}(E \cap K)$ has a finite volume.*

6. *The composition $f \circ g$ is also a piecewise $C^1$-diffeomorphism.*

*Proof. Proof of 1* : Fix $a \in \mathbb{R}^d$. For $x \in \mathbb{R}^d \setminus f(U_f)$, define $f^\dagger(x) = a$, and for $x \in f(V_i)$, define $f^\dagger(x) := f|_{V_i}^{-1}(x)$. Then, $f^\dagger$ is a piecewise $C^1$-mapping with respect to the family of pairwise disjoint open subsets $\{f(V_i)\}_{i \in I}$, and satisfies the conditions for piecewise $C^1$-diffeomphism.

*Proof of 2* : It follows by the following computation:

$$
\int h(x)dx = \int_{f(U_f)} h(x)dx
$$
$$
= \sum_{i \in I} \int_{f(V_i)} h(x)dx
$$
$$
= \sum_{i \in I} \int_{V_i} h(f(x))|Df(x)|dx = \int h(f(x))|Df(x)|dx.
$$

*Proof of 3* It suffices to show that $f^{-1}(K) \cap U_f$ is covered by finitely many compact subsets. In fact, we remark that only finitely many $V_i$'s intersect with $f^{-1}(K)$. If not, infinitely many $f(V_i)$ intersects $f(f^{-1}(K)) \subset K$, which contradicts the definition of piecewise $C^1$-diffeomorphisms. Let $I_0 \subset I$ be a finite subset composed of $i \in I$ such that $V_i$ intersecting with $f^{-1}(K)$. For $i \in I_0$, we define a compact subset $F_i := \tilde{f}_i^{-1}(\tilde{f}_i(\overline{V_i}) \cap K)$. Then we see that $f^{-1}(K) \cap U_f$ is contained in $\cup_{i \in I_0} F_i$.

*Proof of 4* : It suffices to show that for any compact subset $K$, the volume of $f^{-1}(F) \cap K$ is zero. By applying 2 to the case $h = \mathbf{1}_F$, we see that

$$
\int_{f^{-1}(F)} |Df(x)|dx = 0.
$$

For $n > 0$, let $E_n := f^{-1}(F) \cap K \cap \{x \in \mathbb{R}^d : |Df(x)| \geq 1/n\}$. Then we have

$$
\frac{\mathrm{vol}(E_n)}{n} \leq \int_{E_n} |Df(x)|dx \leq \int_{f^{-1}(F)} |Df(x)|dx = 0,
$$

thus $\mathrm{vol}(K \cap f^{-1}(F)) = \lim_{n \to \infty} \mathrm{vol}(E_n) = 0$

*Proof of 5* : By applying 2 to the case $h = \mathbf{1}_{E \cap K}$, we see that

$$
\int_{f^{-1}(E \cap K)} |Df(x)|dx = \mathrm{vol}(E \cap K).
$$

Let $F$ be a closure of $f^{-1}(K) \cap U_f$. By 3, $F$ is a compact subset. Let $I_0 := \{i \in I : F \cap V_i \neq \emptyset\}$ be a finite subset. Then we have

$$C := \inf_{f^{-1}(K) \cap U_f} |Df|$$
$$\geq \inf_{i \in I_0} \inf_{F \cap \overline{V_i}} |D\tilde{f}_i| > 0.$$

Thus,

$$\int_{f^{-1}(E \cap K) \cap U_f} |Df(x)| dx \geq C \operatorname{vol}(f^{-1}(E \cap K)),$$

where the last equality follows from $\operatorname{vol}(f^{-1}(E \cap K) \setminus U_f) = 0$. Thus we have $\operatorname{vol}(f^{-1}(E \cap K)) < \infty$

*Proof of 6 :* We denote by $\{V_i\}_{i \in I}$, $\{V_j'\}_{j \in J}$ the disjoint open families associated with $f$ and $g$, respectively. At first, we prove $f \circ g$ is a piecewise $C^1$-mapping. Let $V_{ij} := g^{-1}(V_i \cap g(V_j')) \cap U_g$ and define $\mathcal{U}_{f \circ g} := \{V_{ij}\}_{(i,j) \in I \times J}$. Let $U_{f \circ g} := \cup_{i,j} V_{ij} = g^{-1}(U_f \cap g(U_g)) \cap U_g$. By 4, the volume of $\mathbb{R}^d \setminus U_{f \circ g}$ is zero. On each $V_{ij}$, $\tilde{f}_i \circ \tilde{g}_j$ is an extension of $f \circ g|_{V_{ij}}$. For any compact subset $K$, $\#\{(i,j) \in I \times J : K \cap V_{ij} \neq \emptyset\} < \infty$. In fact, suppose the number is infinite. Then $g(U_f \cap K)$ intersects with an infinite number of open subsets in the form of $g(U_f \cap K) \cap V_i \cap g(V_j')$. On the other hand $g(U_f \cap K)$ is a bounded subset, thus by definition, the number of $(i,j) \in I \times J$ satisfying $\overline{g(U_f \cap K)} \cap V_i \cap g(V_j') \neq \emptyset$ is finite. It is a contradiction. Therefore, $g \circ f$ is a piecewise $C^1$-mapping.

Next, we prove $f \circ g$ is a piecewise $C^1$-diffeomorphism. The first and second condition follows by definition. For the third condition, since $\mathbb{R}^d \setminus f \circ g(U_{f \circ g}) = (\mathbb{R}^d \setminus f(U_f)) \cup (\mathbb{R}^d \setminus f(g(U_g)) \subset \mathbb{R}^d \setminus f(g(U_g) \cap U_f)$, it suffices to show that the volue of $\mathbb{R}^d \setminus f(g(U_g) \cap U_f)$ is zero. In fact, by the injectivity of $f$ on $U_f$, we have $f(g(U_g) \cap U_f) = f(U_f) \setminus f(U_f \setminus g(U_g))$. Thus $\mathbb{R}^d \setminus f(g(U_g) \cap U_f) = (\mathbb{R}^d \setminus f(U_f)) \cup f(U_f \setminus g(U_g))$. By definition of $C^1$-diffeomorphism, we conclude $\mathbb{R}^d \setminus f(g(U_g) \cap U_f)$ is a nullset. For the fourth condition, let $K$ be a compact subset. Assume the $\{(i,j) \in I \times J : f \circ g(V_{ij}) \cap K \neq \emptyset\} = \infty$. Since $f$ is a piecewise $C^1$-diffeomorphism, there exist infinitely many elements in $j \in J$ such that $f \circ g(V_j') \cap f(U_f) \cap K \neq \emptyset$. On the other hand, $f^{-1}(K \cap f(U_f)) \cap U_f$ is bounded, and its closure intersects with only finitely many $g(V_j')$'s, thus $K \cap f(U_f)$ intersects with only finitely many $f \circ g(V_j')$, which is a contradiction.

$\square$

For a measurable mapping $f : \mathbb{R}^m \to \mathbb{R}^n$ and any $R > 0$, we define a measurable set

$$\mathcal{L}(R; f) := \{x \in \mathbb{R}^m : \|f(x) - f(y)\| > R\|x - y\| \text{ for some } y \in U_f\}.$$

Then we have the following proposition:

**Proposition 5.** *Let $f : \mathbb{R}^m \to \mathbb{R}^n$ be a piecewise $C^1$-mapping. Assume $f$ is linearly increasing, namely, there exists $a, b > 0$ such that $\|f(x)\| < a\|x\| + b$ for any $x \in \mathbb{R}^m$. Then for any compact subset $K'$, $\operatorname{vol}(\mathcal{L}(R; f) \cap K') \to 0$ as $R \to \infty$.*

*Proof.* Let $B$ be an open ball containing $K'$ of radius $r$. Fix an arbitrary $\varepsilon > 0$. We note that the linearly increasing condition implies the locally boundedness of $f$. Let $C := \sup_{\overline{B}} \|f\|$. For $\delta > 0$, we define

$$V_\delta := \{x \in \overline{B} : \operatorname{dist}(x, \partial U_f \cup \partial B)) < \delta\},$$

where $\operatorname{dist}(x, S) := \inf_{y \in S}\{\|x - y\|\}$. Set $\delta$ to be $\operatorname{vol}(V_\delta) < \varepsilon$. We claim that

$$L := \sup_{(x,y) \in K' \times \mathbb{R}^m \setminus B} \frac{\|f(x) - f(y)\|}{\|x - y\|}$$

is finite. In fact, let $r' := \inf_{x \in K', y \notin B} \|x - y\|$. Then for $x \in K'$ and $y \notin B$, we have

$$
\begin{aligned}
\frac{\|f(x) - f(y)\|}{\|x - y\|} &\leq \frac{\|f(x)\| + \|f(y)\|}{\|x - y\|} \\
&\leq \frac{a\|x\| + a\|y\| + 2b}{\|x - y\|} \\
&\leq \frac{a\|x\| + a(\|x - y\| + \|x\|) + 2b}{\|x - y\|} \\
&\leq a + \frac{2a\|x\| + 2b}{\|x - y\|} \\
&< a + \frac{2ar + 2b}{r'}.
\end{aligned}
$$

Thus, $L$ is finite. Since $\overline{B}$ intersects with finitely many $V_i$'s, $f|_{B \setminus V_{\delta/2}}$ is a Lipschitz function. Put $L_\delta > 0$ as the Lipschitz constant of $f|_{B \setminus V_{\delta/2}}$. Then for any $R > \max(L, L_\delta, 4C/\delta)$, we see that $\mathcal{L}(R; f) \cap K'$ is contained in $V_\delta$. Actually, we should prove that $x \notin \mathcal{L}(R; f)$ when $x \in K' \setminus V_\delta$. Take arbitrary $y \in \mathbb{R}^m$. When $y \notin B$, since $x \in K'$, we have $\frac{\|f(x) - f(y)\|}{\|x - y\|} \leq L$ by the definition of $L$. When $y \in B \setminus V_{\delta/2}$, since $x \in K' \setminus V_\delta \subset B \setminus V_{\delta/2}$, we have $\frac{\|f(x) - f(y)\|}{\|x - y\|} \leq L_\delta$ by the definition of $L_\delta$. When $y \in V_{\delta/2}$, we have $\|x - y\| \geq \frac{\delta}{2}$ because $x \notin V_\delta$. Thus,

$$
\frac{\|f(x) - f(y)\|}{\|x - y\|} \leq \frac{\|f(x)\| + \|f(y)\|}{\delta/2} \leq \frac{C + C}{\delta/2} \leq \frac{4C}{\delta}.
$$

Combining these three cases, we conclude that $x \notin \mathcal{L}(R; f)$. Thus we have $\mathrm{vol}(\mathcal{L}(R; f) \cap K') < \varepsilon$, namely, we conclude $\mathrm{vol}(\mathcal{L}(R; f) \cap K') \to 0$ as $R \to \infty$. $\qquad\square$

*Remark.* The linearly increasing condition is important to prove our main theorem. Our approximation targets are compactly supported diffeomorphisms, affine transformations, and the discontinuous ACFs appeared in Section 4.2 or Section D.1, all of which satisfy the linearly increasing condition.

## F    Compatibility of approximation and composition

In this section, we prove the following proposition. It enables the component-wise approximation, i.e., given a transformation that is represented by a composition of some transformations, we can approximate it by approximating each constituent and composing them. The justification of this procedure is not trivial and requires a fine mathematical argument. The results here build on the terminologies and the propositions for piecewise $C^1$-diffeomorphisms presented in Section E.

**Proposition 6.** *Let $\mathcal{M}$ be a set of piecewise $C^1$-diffeomorphisms (resp. locally bounded maps) from $\mathbb{R}^d$ to $\mathbb{R}^d$, and $F_1, \ldots, F_r$ be linearly increasing piecewise $C^1$-diffeomorphisms (resp. continuous maps) from $\mathbb{R}^d$ to $\mathbb{R}^d$ ($r \geq 2$). Assume for any $\varepsilon > 0$ and compact set $K \subset \mathbb{R}^d$, there exists $\widetilde{G}_1, \ldots, \widetilde{G}_r \in \mathcal{M}$ such that for $i \in [r]$, $\left\| F_i - \widetilde{G}_i \right\|_{p,K} < \varepsilon$ (resp. $\left\| F_i - \widetilde{G}_i \right\|_{\mathrm{sup},K} < \varepsilon$). Then for any $\varepsilon > 0$ and compact set $K \subset \mathbb{R}^d$, there exists $G_1, \ldots, G_r \in \mathcal{M}$, such that*

$$
\|F_r \circ \cdots \circ F_1 - G_r \circ \cdots \circ G_1\|_{p,K} < \varepsilon
$$

$$
\left( \textit{resp. } \|F_r \circ \cdots \circ F_1 - G_r \circ \cdots \circ G_1\|_{\mathrm{sup},K} < \varepsilon \right)
$$

*Proof.* We prove by induction. In the case of $r = 2$, it follows by Lemma 15 (for $L^p$-norm) or Lemma 16 (for sup-norm) below in the case of $\mathcal{M}_1 = \mathcal{M}_2 = \mathcal{M}$. In the general case, let $\widetilde{F}_2 := F_r \circ \cdots F_2$. Then by the induction hypothesis, for any compact set $K$ and $\varepsilon > 0$, there exists $\widetilde{G}_2 = G_r \circ \cdots \circ G_2$ for some $G_i \in \mathcal{M}$ such that $\left\| \widetilde{F}_2 - \widetilde{G}_2 \right\|_{?,K} < \varepsilon$, where $? = p$ or sup. By applying Lemma 15 or Lemma 16 with $\mathcal{M}_1 = \mathcal{M}$ and $\mathcal{M}_2 = \mathcal{M} \circ \cdots \circ \mathcal{M}$ (the set of compositions of $r - 1$ elements of $\mathcal{M}$) below, we conclude the proof. $\qquad\square$

**Lemma 15.** *Let $\mathcal{M}_1$ and $\mathcal{M}_2$ be sets of piecewise $C^1$-diffeomorphisms from $\mathbb{R}^d$ to $\mathbb{R}^d$. Let $F_1, F_2 : \mathbb{R}^d \to \mathbb{R}^d$ be linearly increasing piecewise $C^1$-diffeomorphisms. Assume for any $\varepsilon > 0$ and compact set $K \subset \mathbb{R}^d$, for $i = 1, 2$, there exists $\widetilde{G}_i \in \mathcal{M}_i$ such that $\left\| F_i - \widetilde{G}_i \right\|_{p,K} < \varepsilon$. Then for any $\varepsilon > 0$ and compact set $K \subset \mathbb{R}^d$, for $i = 1, 2$, there exists $G_i \in \mathcal{M}_i$, such that*

$$\| F_2 \circ F_1 - G_2 \circ G_1 \|_{p,K} < \varepsilon.$$

*Proof.* Fix arbitrary $\varepsilon > 0$ and compact set $K \subset \mathbb{R}^d$. Put $K' := \overline{F_1(K \cap U_{F_1})}$. Then, since $F_1(K \cap U_{F_1})$ is bounded (see the remark under Definition 12), $K'$ is compact. We claim that there exists $R > 0$ such that

$$\mathrm{vol}(F_1^{-1}\left( \mathcal{L}(R; F_2) \cap K' \right))^{1/p} < \frac{\varepsilon}{3 \mathrm{ess.sup} \|F_2\|},$$

which can be confirmed as follows. Take an increasing sequence $R_n > 0$ ($n \geq 1$) satisfying $\lim_{n \to \infty} R_n = \infty$. Let $B_n := \mathcal{L}(R_n; f) \cap K'$ and $A_n := F_1^{-1}(B_n)$. Then, from Proposition 5, we have $\mathrm{vol}(B_n) \to 0$, which implies $\mathrm{vol}(\bigcap_{n=1}^{\infty} B_n) = 0$. By Proposition 4 (4), we have $\mathrm{vol}(\bigcap_{n=1}^{\infty} A_n) = \mathrm{vol}(F_1^{-1}(\bigcap_{n=1}^{\infty} B_n)) = 0$. By Proposition 4 (5), we have $\mathrm{vol}(A_1) = \mathrm{vol}(F_1^{-1}(B_1)) < \infty$. Recall that if a decreasing sequence $\{S_n\}_{n=1}^{\infty}$ of measurable sets satisfies $\mathrm{vol}(S_1) < \infty$ and $\mathrm{vol}(\bigcap_{n=1}^{\infty} S_n) = 0$, then $\lim_{n \to \infty} \mathrm{vol}(S_n) = 0$. Therefore, we obtain $\lim_{n \to \infty} \mathrm{vol}(A_n) = 0$ and we have the assertion of the claim.

Take $G_1 \in \mathcal{M}_1$ such that

$$\| F_1 - G_1 \|_{p,K} < \frac{\varepsilon}{3R}.$$

Put $S := F_1^{-1}\left( \mathcal{L}(R; F_2) \cap K' \right)$, and define a compact subset $K'' := \overline{(G_1^\dagger)^{-1}(K) \cap U_{G_1^\dagger}}$. Here, the compactness of $K''$ follows from Proposition 4 (3). Next, we take $G_2 \in \mathcal{M}_2$ such that

$$\| F_2 - G_2 \|_{p,K''} < \frac{\varepsilon}{3 \, \underset{(G_1^\dagger)^{-1}(K)}{\mathrm{ess.sup}} \, |\det(DG_1^\dagger)|}$$

where $G_1^\dagger$ is a piecewise $C^1$-diffeomorphism defined by Proposition 4 (1). Then we have

$$\begin{aligned}
& \| F_2 \circ F_1 - G_2 \circ G_1 \|_{p,K} \\
& \leq \| F_2 \circ F_1 - F_2 \circ G_1 \|_{p,K} + \| F_2 \circ G_1 - G_2 \circ G_1 \|_{p,K} \\
& \leq \left\| (F_2 \circ F_1 - F_2 \circ G_1) \mathbf{1}_S \right\|_{p,K} + \left\| (F_2 \circ F_1 - F_2 \circ G_1) \mathbf{1}_{K \setminus S} \right\|_{p,K} \\
& \quad + \underset{(G_1^\dagger)^{-1}(K)}{\mathrm{ess.sup}} |\det(DG_1^\dagger)| \| F_2 - G_2 \|_{p,K''} \\
& < \varepsilon.
\end{aligned}$$

$\square$

**Lemma 16** (compatibility of composition)**.** *Let $\mathcal{M}_1$ and $\mathcal{M}_2$ be sets of locally bounded maps from $\mathbb{R}^d$ to $\mathbb{R}^d$. Let $F_1, F_2 : \mathbb{R}^d \to \mathbb{R}^d$ be continuous maps. Assume for any $\varepsilon > 0$ and compact set $K \subset \mathbb{R}^d$, for $i = 1, 2$, there exists $\widetilde{G}_i \in \mathcal{M}_i$ such that $\left\| F_i - \widetilde{G}_i \right\|_{\mathrm{sup},K} < \varepsilon$. Then for any $\varepsilon > 0$ and compact set $K \subset \mathbb{R}^d$, for $i = 1, 2$, there exists $G_i \in \mathcal{M}_i$, such that*

$$\| F_2 \circ F_1 - G_2 \circ G_1 \|_{\mathrm{sup},K} < \varepsilon.$$

*Proof.* Take any positive number $\epsilon > 0$ and compact set $K \subset \mathbb{R}^d$. Put $r := \max_{k \in K} |F_1(k)|$ and $K' := \{x \in \mathbb{R}^d : |x| \leq r + 1\}$. Let $G_2 \in \mathcal{M}_2$ satisfying

$$\sup_{x \in K'} |F_2(x) - G_2(x)| \leq \frac{\epsilon}{2}.$$

Since any continuous map is uniformly continuous on a compact set, we can take a positive number $\delta > 0$ such that for any $x, y \in K'$ with $|x - y| < \delta$,

$$|F_2(x) - F_2(y)| < \frac{\varepsilon}{2}.$$

From the assumption, we can take $G_1 \in \mathcal{M}_1$ satisfying

$$\sup_{x \in K} |F_1(x) - G_1(x)| \leq \min\{1, \delta\}.$$

Then, it is clear that $F_1(K) \subset K'$ by the definition of $K'$. Moreover, we have $G_1(K) \subset K'$. In fact, we have

$$|G_1(k)| \leq \sup_{x \in K} |F_1(x) - G_1(x)| + |F_2(k)| \leq 1 + r \quad (k \in K).$$

Then for any $x \in K$, we have

$$|F_2 \circ F_1(x) - G_2 \circ G_1(x)| \leq |F_2(F_1(x)) - F_2(G_1(x))| + |F_2(G_1(x)) - G_2(G_1(x))|$$
$$< \epsilon.$$

$\square$

# G   Examples of flow architectures covered in this paper

Here, we provide the proofs for the universal approximation properties of certain CF-INNs.

## G.1   Neural autoregressive flows (NAFs)

In this section, we prove that *neural autoregressive flows* [18] yield sup-universal approximators for $\mathcal{S}_c^1$ (hence for $\mathcal{S}_c^\infty$). The proof is not merely an application of a known result in Huang et al. [48] but it requires additional non-trivial consideration to enable the adoption of Lemma 3 in Huang et al. [48] as it is applicable only for those smooth mappings that match certain boundary conditions.

**Definition 14.** A *deep sigmoidal flow* (DSF; a special case of neural autoregressive flows) [18, Equation (8)] is a flow layer $g = (g_1, \ldots, g_d) \colon \mathbb{R}^d \to \mathbb{R}^d$ of the following form:

$$g_k(\boldsymbol{x}) := \sigma^{-1} \left( \sum_{j=1}^n w_{k,j}(\boldsymbol{x}_{\leq k-1}) \cdot \sigma \left( \frac{x_k - b_{k,j}(\boldsymbol{x}_{\leq k-1})}{\tau_j(\boldsymbol{x}_{\leq k-1})} \right) \right),$$

where $\sigma$ is the sigmoid function, $n \in \mathbb{N}$, $w_j, b_j, \tau_j \colon \mathbb{R}^{k-1} \to \mathbb{R}$ $(j \in [n])$ are neural networks such that $b_j(\cdot) \in (r_0, r_1)$, $\tau_j(\cdot) \in (0, r_2)$, $w_j(\cdot) > 0$, and $\sum_{j=1}^n w_j(\cdot) = 1$ $(r_0, r_1 \in \mathbb{R}, r_2 > 0)$. We define DSF to be the set of all possible DSFs.

**Proposition 7.** *The elements of* DSF *are locally bounded, and* $\mathrm{INN}_{\mathrm{DSF}}$ *is a* sup-*universal approximator for* $\mathcal{S}_c^1$.

*Proof.* The elements of DSF are continuous, hence locally bounded. Let $s = (s_1, \cdots, s_d) \in \mathcal{S}_c^1$. Take any compact set $K \subset \mathbb{R}^d$ and $\epsilon > 0$. Since $K$ is compact, there exist $r_0, r_1 \in \mathbb{R}$ such that $K \subset [r_0, r_1]^d$. Put $r_0' = r_0 - 1, r_1' = r_1 + 1$. We take a $C^1$-function $b \colon (r_0', r_1') \to \mathbb{R}$ satisfying

1. $b|_{[r_0, r_1]} = 0$,

2. $b|_{(r_0', r_0)}$ and $b|_{(r_1, r_1')}$ are strictly increasing,

3. $\lim_{x \to r_0' + 0} b(x) = -\infty$ and $\lim_{x \to r_1' - 0} b(x) = \infty$,

4. $\lim_{x \to r_0' + 0} \frac{d(\sigma \circ b)}{dx}(x)$ and $\lim_{x \to r_1' - 0} \frac{d(\sigma \circ b)}{dx}(x)$ exist in $\mathbb{R}$,

where $\sigma$ is the sigmoid function. For each $k \in [d]$, we define a $C^1$-map $\tilde{s}_k \colon [r_0', r_1']^{k-1} \times (r_0', r_1') \times [r_0', r_1']^{d-k} \to \mathbb{R}$, which is strictly increasing with respect to $x_k$, by

$$\tilde{s}_k(x) := s_k(x) + b(x_k) \quad (x = (x_1, \cdots, x_d)).$$

Moreover, we define a map $S\colon [r_0', r_1']^d \to [0,1]^d$ by

$$S_k|_{[r_0', r_1']^{k-1} \times (r_0', r_1') \times [r_0', r_1']^{d-k}} = \sigma \circ \tilde{s}_k,$$
$$S_k(x_1, \cdots, x_{k-1}, r_0', x_{k+1}, \cdots, x_d) = 0,$$
$$S_k(x_1, \cdots, x_{k-1}, r_1', x_{k+1}, \cdots, x_d) = 1,$$

where we write $S = (S_1, \cdots, S_d)$. Then, by Lemma 17, $S$ satisfies the assumptions of Lemma 3 in [48]. Since $S([r_0, r_1]^d) \subset (0,1)^d$ is compact, there exists a positive number $\delta > 0$ such that

$$S([r_0, r_1]^d) + B(\delta) := \{S(x) + v \; : \; x \in [r_0, r_1]^d, v \in B(\delta)\} \subset [\delta, 1-\delta]^d,$$

where $B(\delta) := \{x \in \mathbb{R}^d : |x| \le \delta\}$. Let $L > 0$ be a Lipschitz constant of $\sigma^{-1}\colon (0,1)^d \to \mathbb{R}^d$ on $[\delta, 1-\delta]^d$. By Lemma 3 in [48], there exists $g \in \mathrm{INN}_{\mathrm{DSF}}$ such that

$$\|S - \sigma \circ g\|_{\sup, [r_0', r_1']^d} < \min\left\{\delta, \frac{\epsilon}{L}\right\}.$$

As a result, $\sigma \circ g([r_0, r_1]^d) \subset S([r_0, r_1]^d) + B(\delta) \subset [\delta, 1-\delta]^d$. Then we obtain

$$
\begin{aligned}
\|s - g\|_{\sup, K} \le \|s - g\|_{\sup, [r_0, r_1]^d} &= \|\sigma^{-1} \circ \sigma \circ s - \sigma^{-1} \circ \sigma \circ g\|_{\sup, [r_0, r_1]^d} \\
&\le L \|S - \sigma \circ g\|_{\sup, [r_0, r_1]^d} \\
&< \epsilon.
\end{aligned}
$$

$\square$

**Lemma 17.** *We denote by $\mathcal{T}^1$ the set of all $C^1$-increasing triangular mappings from $\mathbb{R}^d$ to $\mathbb{R}^d$. For $s = (s_1, \cdots, s_d) \in \mathcal{T}^1$, we define a map $S\colon [r_0', r_1']^d \to [0,1]^d$ as in the proof of Proposition 7. Then $S$ is a $C^1$-map.*

*Proof.* It is enough to show that $S_d\colon [r_0', r_1']^d \to [0,1]$ is a $C^1$-function. We prove that for any $i \in [d]$, the $i$-th partial derivative of $S_d$ exists and that it is continuous on $[r_0', r_1']^d$. First, for $i \in [d-1]$, we consider the $i$-th partial derivative.
**Claim 1.**

$$\frac{\partial S_d}{\partial x_i}(x) = \begin{cases} \frac{d\sigma}{dx}(s_i(x) + b(x_d)) \frac{\partial s_d}{\partial x_i}(x) & (x \in [r_0', r_1']^{d-1} \times (r_0', r_1')) \\ 0 & (x_d = r_0', r_1') \end{cases}$$

In fact, for $x \in [r_0', r_1']^{d-1} \times (r_0', r_1')$, we have

$$\frac{\partial S_d}{\partial x_i}(x) = \frac{\partial(\sigma \circ \tilde{s}_d)}{\partial x_i}(x) = \frac{d\sigma}{dx}(s_d(x) + b(x_d))\left(\frac{\partial s_d}{\partial x_i}(x) + 0\right).$$

For $x = (x_{\le d-1}, r_0')$, we have

$$
\begin{aligned}
\frac{\partial S_d}{\partial x_i}(x) &= \lim_{h \to 0} \frac{S_d(x_{\le i-1}, x_i + h, x_{i+1}, \cdots, x_{d-1}, r_0') - S_d(x_{\le d-1}, r_0')}{h} \\
&= \lim_{h \to 0} \frac{0 - 0}{h} = 0
\end{aligned}
$$

Here, note that by the definition of $S_d$, the notation $S_d(x_{\le i-1}, x_i + h, x_{i+1}, \cdots, x_{d-1}, r_0')$ makes sense even if $x_i = r_0'$ or $x_i = r_1'$. We can verify the case $x = (x_{\le d-1}, r_1')$ similarly.

Next, we show that $\frac{\partial S_d}{\partial x_i}$ is continuous. We take any $x_{\le d-1} \in [r_0', r_1']^{d-1}$. Since we have $\lim_{x \to r_0'} b(x) = -\infty$, $\lim_{x \to r_1'} b(x)$, $\lim_{x \to \pm\infty} \frac{d\sigma}{dx}(x) = 0$, and $|\frac{\partial s_d}{\partial x_I}(x)| < \infty \; (x \in [r_0', r_1']^d)$, we obtain

$$\lim_{x \to (x_{d-1}, r_0')} \frac{d\sigma}{dx}(s_i(x) + b(x_d)) \frac{\partial s_d}{\partial x_i}(x) = 0,$$

$$\lim_{x \to (x_{d-1}, r_1')} \frac{d\sigma}{dx}(s_i(x) + b(x_d)) \frac{\partial s_d}{\partial x_i}(x) = 0.$$

Therefore, the partial derivative $\frac{\partial S_d}{\partial x_i}(x)$ is continuous on $[r_0', r_1']^d$ for $i \in [d-1]$.

Next, we consider the $d$-th derivative of $S_d$.

**Claim 2.**

$$\frac{\partial S_d}{\partial x_d}(x) = \begin{cases} \frac{d\sigma}{dx}(s_d(x) + b(x_d))\left(\frac{\partial s_d}{\partial x_d}(x) + \frac{db}{dx}(x_d)\right) & (x \in [r_0', r_1']^{d-1} \times (r_0', r_1')) \\ e^{s_d(x_{\leq d-1}, r_0')}\lim_{x \to r_0'+0}\frac{d(\sigma \circ b)}{dx}(x) & (x_d = r_0') \\ e^{-s_d(x_{\leq d-1}, r_1')}\lim_{x \to r_1'-0}\frac{d(\sigma \circ b)}{dx}(x) & (x_d = r_1') \end{cases}$$

We verify Claim 2. Since it is clear for the case $x \in [r_0', r_1']^{d-1} \times (r_0', r_1')$ by the definition of $S_k$, we consider the case $x_d = r_0', r_1'$.

**Subclaim.** For $x_{\leq d-1}' \in [r_0', r_1']^{d-1}$,

$$\lim_{x \to (x_{\leq d-1}', r_0')}\frac{\sigma(s_d(x) + b(x_d))}{\sigma(b(x_d))} = e^{s_d(x_{\leq d-1}', r_0')}$$

$$\lim_{x \to (x_{\leq d-1}', r_1')}\frac{\sigma(s_d(x) + b(x_d)) - 1}{\sigma(b(x_d)) - 1} = e^{-s_d(x_{\leq d-1}', r_1')}$$

We verify this subclaim. From $\lim_{x \to r_0'} b(x) = -\infty$, we have

$$\frac{\sigma(s_d(x) + b(x_d))}{\sigma(b(x_d))} = \frac{1 + e^{-b(x_d)}}{1 + e^{-s_d(x) - b(x_d)}} = \frac{e^{b(x_d)} + 1}{e^{b(x_d)} + e^{-s_d(x)}}$$

$$\to \frac{1}{e^{-s_d(x_{\leq d-1}', r_0')}} = e^{s_d(x_{\leq d-1}', r_0')} \quad (x \to (x_{\leq d-1}', r_0'))$$

Similarly, from $\lim_{x \to r_1'} b(x) = \infty$, we have

$$\frac{\sigma(s_d(x) + b(x_d)) - 1}{\sigma(b(x_d)) - 1} = e^{-s_d(x)}\frac{1 + e^{-b(x_d)}}{1 + e^{-s_d(x) - b(x_d)}}$$

$$\to e^{-s_d(x_{\leq d-1}', r_1')} \quad (x \to (x_{\leq d-1}', r_1')).$$

Therefore, our subclaim has been proved. By using L'Hôpital's rule, we have

$$\lim_{h \to +0}\frac{\sigma(b(r_0' + h))}{h} = \lim_{x \to r_0'}\frac{d(\sigma \circ b)}{dx}(x), \quad \lim_{x \to r_1'}\frac{\sigma(b(r_1' + h)) - 1}{h} = \lim_{x \to r_1'}\frac{d(\sigma \circ b)}{dx}(x).$$

Then, from Subclaim, we obtain

$$\frac{\partial S_d}{\partial x_d}(x_{\leq d-1}, r_0') = \lim_{h \to +0}\frac{\sigma(s_d(x_{\leq d-1}, r_0' + h) + b(r_0' + h)) - 0}{h}$$

$$= \lim_{h \to +0}\frac{\sigma(s_d(x_{\leq d-1}, r_0' + h) + b(r_0' + h))}{\sigma(b(r_0 + h))} \cdot \frac{\sigma(b(r_0' + h))}{h}$$

$$= e^{s_d(x_{\leq d-1}, r_0')}\lim_{x \to r_0'+0}\frac{d(\sigma \circ b)}{dx}(x),$$

$$\frac{\partial S_d}{\partial x_d}(x_{\leq d-1}, r_1') = \lim_{h \to -0}\frac{\sigma(s_d(x_{\leq d-1}, r_1' + h) + b(r_1' + h)) - 1}{h}$$

$$= \lim_{h \to -0}\frac{\sigma(s_d(x_{\leq d-1}, r_1' + h) + b(r_1' + h)) - 1}{\sigma(b(r_1' + h)) - 1} \cdot \frac{\sigma(b(r_1' + h)) - 1}{h}$$

$$= e^{s_d(x_{\leq d-1}, r_1')}\lim_{x \to r_1'}\frac{d(\sigma \circ b)}{dx}(x).$$

Therefore, Claim 2 was proved.

Finally, we verify $\frac{\partial S_d}{\partial x_d}(x)$ is continuous on $[r_0', r_1']^d$. Fix $x_{\leq d-1}' \in [r_0', r_1']^{d-1}$. Since we have $\lim_{x \to (x_{\leq d-1}', r_0')}\frac{d\sigma}{dx}(\sigma_d(x) + b(x_d))\frac{\partial s_d}{\partial x_d}(x) = 0$, from Claim 2, it is enough to show the following:

**Claim 3.**

$$\lim_{x \to (x_{\leq d-1}', r_0')}\frac{d\sigma}{dx}(s_d(x) + b(x_d))\frac{db}{dx}(x_d) = e^{s_d(x_{\leq d-1}, r_0')}\lim_{x \to r_0'+0}\frac{d(\sigma \circ b)}{dx}(x),$$

$$\lim_{x \to (x_{\leq d-1}', r_1')}\frac{d\sigma}{dx}(s_d(x) + b(x_d))\frac{db}{dx}(x_d) = e^{-s_d(x_{\leq d-1}, r_1')}\lim_{x \to r_1'-0}\frac{d(\sigma \circ b)}{dx}(x).$$

We verify Claim 3. We have

$$\frac{d\sigma}{dx}(s_d(x) + b(x_d))\frac{db}{dx}(x_d) = \frac{\frac{d\sigma}{dx}(s_d(x) + b(x_d))}{\frac{d\sigma}{dx}(b(x_d))}\frac{d\sigma}{dx}(b(x_d))\frac{db}{dx}(x_d)$$

$$= \frac{\frac{d\sigma}{dx}(s_d(x) + b(x_d))}{\frac{d\sigma}{dx}(b(x_d))}\frac{d(\sigma \circ b)}{dx}(x_d).$$

Since we have $\frac{d\sigma}{dx}(x) = \sigma(x)(1 - \sigma(x))$, from Subclaim above, Claim 3 follows from

$$\frac{\frac{d\sigma}{dx}(s_d(x) + b(x_d))}{\frac{d\sigma}{dx}(b(x_d))} = \frac{\sigma(s_d(x) + b(x_d))}{\sigma(b(x_d))} \cdot \frac{1 - \sigma(s_d(x) + b(x_d))}{1 - \sigma(b(x_d))}$$

$$\rightarrow \begin{cases} e^{s_d(x'_{\leq d-1}, r'_0)} & (x \rightarrow (x'_{\leq d-1}, r'_0)) \\ e^{-s_d(x'_{\leq d-1}, r'_1)} & (x \rightarrow (x'_{\leq d-1}, r'_1)) \end{cases}.$$

Therefore, we proved the continuity of $\frac{\partial S_d}{\partial x_d}(x)$. $\qquad\square$

## G.2 Sum-of-squares polynomial flows (SoS flows)

In this section, we prove that *sum-of-squares polynomial flows* [21] yield CF-INNs with the sup-universal approximation property for $\mathcal{S}_c^1$ (hence for $\mathcal{S}_c^\infty$). Even though Jaini et al. [21] claimed the distributional universality of the SoS flows by providing a proof sketch based on the univariate Stone-Weierstrass approximation theorem, we regard the sketch to be invalid or at least incomplete as it does not discuss the smoothness of the coefficients, i.e., whether the polynomial coefficients can be realized by continuous functions. Here, we provide complete proof that takes an alternative route to prove the sup-universality of the SoS flows via the multivariate Stone-Weierstrass approximation theorem.

**Definition 15.** A *sum-of-squares polynomial flow* (SoS flow) [21, Equation (9)] is a flow layer $g = (g_1, \ldots, g_d)\colon \mathbb{R}^d \rightarrow \mathbb{R}^d$ of the following form:

$$g_k(\boldsymbol{x}) := \mathfrak{B}_{2r+1}(x_k; C_k(\boldsymbol{x}_{\leq k-1})),$$

$$\mathfrak{B}_{2r+1}(z; (c, \boldsymbol{a})) := c + \int_0^z \sum_{b=1}^B \left(\sum_{l=0}^r a_{l,b}u^l\right)^2 du,$$

where $C_k\colon \mathbb{R}^{k-1} \rightarrow \mathbb{R}^{B(r+1)+1}$ is a neural network, $r \in \mathbb{N} \cup \{0\}$, and $B \in \mathbb{N}$. We define SoS to be the set of all possible SoS flows.

**Proposition 8.** *The elements of* SoS *are locally bounded, and* $\mathrm{INN}_{\mathrm{SoS}}$ *is a* sup-*universal approximator for* $\mathcal{S}_c^1$.

*Proof.* The elements of SoS are continuous, hence locally bounded. The sup-universality follows from the Stone-Weierstrass approximation theorem as in the below. Let $s = (s_1, \ldots, s_d) \in \mathcal{S}_c^1$, a compact subset $K \subset \mathbb{R}^d$, and $\epsilon > 0$ be given. Then, there exists $R > 0$ such that $K \subset [-R, R]^d$. Since $s_d(\boldsymbol{x})$ is strictly increasing with respect to $x_d$ and $s$ is $C^1$, we have $\eta(\boldsymbol{x}) := \frac{\partial s_d}{\partial x_d}(\boldsymbol{x}) > 0$ and $\eta$ is continuous. Therefore, we can apply the Stone-Weierstrass approximation theorem [46, Corollary 4.50] to $\sqrt{\eta(\boldsymbol{x})}$: for any $\delta > 0$, there exists a polynomial $\pi(x_1, \ldots, x_d)$ such that $\left\|\sqrt{\eta} - \pi\right\|_{\mathrm{sup},[-R,R]^d} < \delta$. Then, by rearranging the terms, there exist $r \in \mathbb{N}$ and polynomials $\xi_l(x_1, \ldots, x_{d-1})$ such that $\pi(x_1, \ldots, x_d) = \sum_{l=0}^r \xi_l(x_1, \ldots, x_{d-1})x_d^l$. Now, define

$$\tilde{g}_d(\boldsymbol{x}) := s_d(\boldsymbol{x}_{\leq d-1}, 0) + \int_0^{x_d} (\pi(\boldsymbol{x}_{\leq d-1}, u))^2 du$$

$$= s_d(\boldsymbol{x}_{\leq d-1}, 0) + \int_0^{x_d} \left(\sum_{l=0}^r \xi_l(x_1, \ldots, x_{d-1})u^l\right)^2 du$$

and $\tilde{g}(\boldsymbol{x}) := (x_1, \ldots, x_{d-1}, \tilde{g}_d(\boldsymbol{x}))$. Then,

$$
\begin{aligned}
\|s - \tilde{g}\|_{\sup, K} &= \sup_{\boldsymbol{x} \in K} |s_d(\boldsymbol{x}) - \tilde{g}_d(\boldsymbol{x})| \\
&= \sup_{\boldsymbol{x} \in K} \left| s_d(\boldsymbol{x}_{\leq d-1}, 0) + \int_0^{x_d} \eta(\boldsymbol{x}_{\leq d-1}, u) du - \tilde{g}_d(\boldsymbol{x}) \right| \\
&= \sup_{\boldsymbol{x} \in K} \left| \int_0^{x_d} (\sqrt{\eta(\boldsymbol{x}_{\leq d-1}, u)}^2 - \pi(\boldsymbol{x}_{\leq d-1}, u)^2) du \right| \\
&\leq R \cdot \sup_{\boldsymbol{x} \in [-R,R]^d} \left| \sqrt{\eta(\boldsymbol{x})}^2 - \pi(\boldsymbol{x})^2 \right| \\
&= R \cdot \sup_{\boldsymbol{x} \in [-R,R]^d} |\sqrt{\eta(\boldsymbol{x})} + \pi(\boldsymbol{x})| \cdot |\sqrt{\eta(\boldsymbol{x})} - \pi(\boldsymbol{x})| \\
&\leq R \left( \sup_{\boldsymbol{x} \in [-R,R]^d} 2\sqrt{\eta(\boldsymbol{x})} + \delta \right) \delta,
\end{aligned}
$$

where we used

$$
\begin{aligned}
\sup_{\boldsymbol{x} \in [-R,R]^d} |\sqrt{\eta(\boldsymbol{x})} + \pi(\boldsymbol{x})| &\leq \sup_{\boldsymbol{x} \in [-R,R]^d} |2\sqrt{\eta(\boldsymbol{x})}| + |\sqrt{\eta(\boldsymbol{x})} - \pi(\boldsymbol{x})| \\
&\leq \sup_{\boldsymbol{x} \in [-R,R]^d} 2\sqrt{\eta(\boldsymbol{x})} + \delta.
\end{aligned}
$$

It is straightforward to show that there exists $g \in \mathrm{SoS}$ such that $\|\tilde{g} - g\|_{\sup, K} < \frac{\epsilon}{2}$ by approximating each of $s_d(\boldsymbol{x}_{\leq d-1})$ and $\xi_l$ on $K$ using neural networks. Finally, taking $\delta$ to be small enough so that $\|s - \tilde{g}\|_{\sup, K} < \frac{\epsilon}{2}$ holds, the assertion is proved. $\qquad \square$

## H  Using permutation matrices instead of $\mathrm{Aff}$ in the definition of $\mathrm{INN}_{\mathcal{G}}$

In terms of representation power, there is no essential difference between using the permutation group and using the general linear group in Definition 1. In fact, one can express the elementary operation matrices (hence the regular matrices) by combining affine coupling flows, permutations.

From this result, we can see that employing $\mathrm{Aff}$ in Definition 1 instead of the permutation matrices is not an essential requirement for the universal approximation properties to hold. For this reason, we believe that the empirically reported difference in the performances of Glow [4] and RealNVP [3] is mainly in the efficiency of approximation rather than the capability of approximation.

**Lemma 18.** *We have*

$$
\mathrm{INN}_{\mathcal{H}\text{-ACF}} = \{W_1 \circ g_1 \circ \cdots \circ W_n \circ g_n \;:\; g_i \in \mathcal{H}\text{-ACF}, W_i \in \mathfrak{S}_d\}, \tag{1}
$$

*where $\mathfrak{S}_d$ is the permutation group of degree $d$.*

*Proof.* Since any translation operator (i.e., addition of a constant vector) can be easily represented by the elements of $\mathcal{H}$-ACF and permutations, it is enough to show that any element of $\mathrm{GL}(n, \mathbb{R})$ can be realized by a finite composition of elements of $\mathcal{H}$-ACF and $\mathfrak{S}_d$. To show that, it is sufficient to consider only the elementary matrices. Row switching comes from $\mathfrak{S}_d$. Moreover, element-wise sign flipping can be described by a composition of finite elements of $\mathcal{H}$-ACF. To see this, first observe that

$$
\begin{pmatrix} -1 & 0 \\ 0 & 1 \end{pmatrix} = \begin{pmatrix} 1 & 0 \\ 1 & 1 \end{pmatrix} \begin{pmatrix} 0 & 1 \\ 1 & 0 \end{pmatrix} \begin{pmatrix} 1 & 0 \\ -1 & 1 \end{pmatrix} \begin{pmatrix} 0 & 1 \\ 1 & 0 \end{pmatrix} \begin{pmatrix} 1 & 0 \\ 1 & 1 \end{pmatrix} \begin{pmatrix} 0 & 1 \\ 1 & 0 \end{pmatrix}
$$

holds. Now, any lower triangular matrix with positive diagonals can be described by a composition of finite elements of $\mathcal{H}$-ACF. Therefore, any diagonal matrix whose components are $\pm 1$ can be described by a composition of elements in $\mathcal{H}$-ACF and $\mathfrak{S}_d$. Therefore, any affine transform is an element of the right hand side of (1). $\qquad \square$

## I  Other related work

In this section, we elaborate on the relation of the present paper and the existing literature.

**Approach to make universal approximators by augmenting the dimensionality.** Zhang et al. [39] showed that invertible residual networks (i-ResNets) [49] and neural ordinary differential equations (NODEs) [37, 38] can be turned into universal approximators of homeomorphisms by increasing the dimensionality and padding zeros. Similarly, Huang et al. [50] motivated employing the dimensionality augmentation technique for ACFs based on the theory of Hamiltonian ODEs. Given that, one may wonder if we can apply a similar technique to augment CF-INN to have the universality, which can bypass the proof techniques developed in this study. However, there is a problem that the approach can undermine the exact invertibility of the model: unless the model is ideally trained so that it always outputs zeros in the zero-padded dimensions, the model can no longer represent an invertible map operating on the original dimensionality. On the other hand, we showed the universality properties of certain CF-INNs without introducing the complication arising from the dimensionality augmentation.