[Reviews · NeurIPS 2020]

Review 1

Summary and Contributions: The paper proves multiple coupling based invertible nets can universally approximate smooth diffeomorphisms, and as a byproduct establishes the equivalence of single-coordinate transforms, triangular transforms and C^2 diffeomorphisms through function composition, which is a somewhat surprising and stronger result than distributional universality commonly seen in the literature.

Strengths: The results proven in this work do not seem trivial, and can potentially help practitioners gain a better intuition to improve the parameterization of invertible neural networks, especially because coupling-based INN is the most commonly adopted architecture.

Weaknesses: The need of having a diffeomorphic universality for discrete time NF is not well motivated, asides from the fact that it implies distributional universality. On what other occasion would this result be useful?

Correctness: I didn’t go through the appendix to read the proofs in detail, but the sketch of proof presented in the main text seems convincing and reasonable.

Clarity: The paper is well written and easy to follow on the whole.

Relation to Prior Work: The following refs are missing. [1] attempts to prove Real-NVP type of coupling flow are distributionally universal. [2] also introduces universal non-linear 1D transformation via spline. [3] discusses the expressivity power of planar and sylvester (perhaps not as relevant since it’s not the same type of flow). [1] Solving ODE with Universal Flows: Approximation Theory for Flow-Based Models [2] Neural Spline Flows [3] The Expressive Power of a Class of Normalizing Flow Models

Reproducibility: Yes

Additional Feedback: [POST REBUTTAL] --------- I thank the authors for the detailed response. I guess I see how one can parameterize some (Real NVP-style) linear couplings combined with permutation and sign flipping to represent any regular matrices (regular here denotes invertible I suppose?). Perhaps this could be explicitly constructed in the paper to complement the results. Does it also imply the the general linear group in the main result can be replaced with permutation group + sign flipping? Furthermore, if someone is only concerned with diffeomorphisms with a jacobian having strictly positive eigenvalues, then can the sign flipping be dropped? I raised my overall rating to 8 due to the non-triviality of the results presented in the paper, and the clear exposition. I hope the authors can take into account the feedback/additional questions to update the paper, and I look forward to reading the updated version. --------- Can you comment on the argument of [4], and whether it contradicts the results presented in this work. One key difference that I observe is that INN_H-ACF is intertwined with linear maps whereas regular Real-NVP type of coupling is followed by permutation. This perhaps emphasizes the importance in linear maps (aside from the fact that it allows one to flip the direction). Can you point out the use of this linear map in establishing diffeomorphic universality? Perhaps this explains the empirical advantage of Glow over RealNVP as it generalizes the permutation to linear maps (albeit 1x1). Also, I think it’s not accurate enough to directly say (in line 191) Glow is universal since the family of linear maps it uses is limited (so is the ACF class, since it’s convolutional). [4] You say Normalizing Flows I see Bayesian Networks


Review 2

Summary and Contributions: The work provides several theoretical results concerning the universality of coupling-based invertible neural networks (CF-INNs). The central contribution is the rigorous proof that affine coupling flows (ACFs), the most widely used architecture in current research, is distributional- and Lp-universal.

Strengths: To the INN- and NF-community, I see this work having a tremendous significance, and being an important missing ingredient in bringing the field of NFs forward. Firstly, it addresses an important and long-standing question: Specialized architectures such as NAF and SoS flows have been shown to be distributional universal approximators in the past. But in reality, the majority of practical works use affine coupling flows (ACFs), where distributional universality had not been shown. But this work goes above and beyond, and even shows that they are Lp-universal Diffeomorphism approximators, which is a much stronger property, and relevant for a broader range of works. Secondly, the results are formulated and presented in such a way that make them easy for others in the community to use: The work shows various connections and implications between different notions of universality and architectures, e.g. Theorem 1, Lemma 1, lines 186-192, etc. This is extremely useful for the future, in proving the reliability of newly proposed methods in different contexts. These extra results are stated clearly and provided in a helpful and accessible manner.

Weaknesses: I do not see a weakness concerning the content itself or the methodology.

Correctness: The proofs are carefully and rigorously conduced, having read most of the appendix. However, I am no expert in differential geometry, and I did not myself verify that the conditions for all cited theorems that are used were met (therefore my confidence score of 3).

Clarity: While the paper is quite technical, the language and notation is very precise and well thought-out, avoiding most misunderstandings. Furthermore, I think dividing the main paper into "Main Results" (Sec 3) and "Proof Outline" (Sec 4) is a good idea. Section 3 is compact and simple enough that more practically oriented researchers should be able to understand and cite the results correctly. Section 4 explains the main ideas of the proofs and illustrates them with figures. The figures are helpful, and most explanations are easy to follow. However, I do think lines 212-235 could be more straightforward; I had to read this twice to understand all the reasoning and look at parts of the appendix in between. Perhaps the arguments and explanations could be sorted in a different order. E.g. flow endpoints are introduced, but it is not clear why this is useful until l.227. Instead, a sentence could be spent around line 216, saying that the "additivity" of flow endpoints will later be useful to decompose a non-nearly-id map into a composition of identical nearly-id maps.

Relation to Prior Work: The relevant related work is discussed. Some related work is also extended upon, see line 162ff. A remark that does not influence my score: For the related work, I think some works from Bayesian inference could be relevant, for instance Bigoni et al, "Greedy inference with layers of lazy maps". While the nomenclature is different, the 'Lazy maps' are essentially also CF-INNs, and they show some guarantees in terms of KL-divergence.

Reproducibility: Yes

Additional Feedback: ================ Update after rebuttal ================ Some additional criticisms were brought up by the other reviewers. The ones I found most relevant were also addressed properly in the rebuttal. I leave my score unchanged.


Review 3

Summary and Contributions: The paper shows universality results for affine coupling flows for invertible function approximation. ******Post Rebuttal********** Thank you for the response. My comment about the work not being quantitative was indeed not meant as a major criticism and this being one of the first works on the problem it addresses, I agree with the authors that this is a good first step.

Strengths: The results are clean and the proof techniques based on some basic theorems from differential topology/geometry seem new. I found the results somewhat surprising

Weaknesses: My main concern is that the results here are not "quantitative": e.g. they does not show what depth r (number of composition) is required to achieve error \epsilon. It is known that in certain cases this can be an issue: [The Expressive Power of a Class of Normalizing Flow Models, Zhifeng Kong, Kamalika Chaudhuri, https://arxiv.org/abs/2006.00392]. Thus, even if the class of functions is universal in the sense of the present paper, it may not be so in practice where the depth r may be a relatively small number (say 100). This sort of limitation is also shared by various classical universality results such as Cybenko's.

Correctness: Proofs seem correct though I did not check all the details.

Clarity: Well written.

Relation to Prior Work: Well done (but see the reference above).

Reproducibility: Yes

Additional Feedback: Marther --> Mather (Line 221)


Review 4

Summary and Contributions: In this paper, the author provides rigorous proof of a proposed theorem to show the universal approximation properties for certain types of functions. This will provide the guarantee of universality of a CF-INN. This paper also provides a solid answer t othe distributional universality of ACF-based CF-INNs.

Strengths: The claims in this paper are rigorously proved in view of differential geometry. The results provide the theoretical understanding of a CF-INNs by illustrating the universal approximation. Furthermore, it provides some deeper insight on the functionality of different flow layer designs in the model class.

Weaknesses: As a theoretical paper, it focuses on illustrating the universal approximation on a special type of INN, thus the contribution is limited. No experimental results are presented to demonstrate how the theory works on existing algorithms.

Correctness: The claims in this paper are correct and the proofs are rigorous and elegant.

Clarity: The paper is well structured and well written.

Relation to Prior Work: Due to the novelty of this topic, the related literature is limited. This paper advances the theoretical understanding of previous work.

Reproducibility: Yes

Additional Feedback: [POST REBUTTAL] I thank the authors for the detailed response, which address all of my concerns. I carefully check the proofs in the supplementary materials, and am convinced. Hence I change to my score to 7 accept. It will be helpful to further discuss the following problem: current method aims at using a sequence of diffeomorphisms to map one probability measure to the other one, this can be achieved more directly using rigorous optimal tansportation method. The comparison between these two approaches will be helpful. The second issue is the choice of function famility as the approximator, why the current one is the most appropriate one to reflect the reality?

[Author Response · NeurIPS 2020]

**Re: Coupling-based Invertible Neural Networks Are Universal Diffeomorphism Approximators (ID=1064).**

We thank the reviewers for reviewing our work. We will update the paper based on the suggestions.

**To Reviewer #1**

**Q1-1.** On what occasion would the diffeomorphic universality results be useful other than distribution approximation?

**A1-1.** As explained in lines 29–33 of the paper, we believe the result is relevant when these INNs are used to learn an
invertible transformation, e.g., in feature extraction or independent component analysis.

**Q1-2.** Missing references [1, 2, 3].

**A1-2.** Thank you for pointing out the missing references. We will include them in an updated version: as for [1] and
[3], we will explain the relation in Supplementary H. As for [2], we will introduce it in Section 5.2.

**Q1-3.** Can you comment on the argument of [4], and whether it contradicts the results presented in this work?

**A1-3.** We first note that it is hard to technically verify whether the argument of [4] is complete as it is not rigorously
stated. That said, to our best understanding, the argument of [4] does not properly take into account the approximation
perspective hence fails to prove the non-universality. More concretely, the counterexample proposed in [4] critically
relies on the independence of the target distribution. However, as far as approximation is concerned, non-independent
distributions can approximate independent distributions. Specifically, our Lemma 12 (Supplementary F) seems to
circumvent the first case of the contradiction argument of [4] because Lemma 12 shows that we can approximate a
nonlinear component-wise transformation by using affine coupling flows.

**Q1-4.** Does the difference between the permutation layers and the invertible linear layers essentially contribute to the
diffeomorphism universality? Does this result imply that Glow has superior representation power over RealNVP?

**A1-4.** In terms of representation power, the difference between using the permutation group and using the general
linear group is relatively small: as small as component-wise sign swapping (i.e., a layer to multiply some dimensions
by $-1$). In fact, one can express the elementary operation matrices (hence the regular matrices) by combining affine
coupling flows, permutations, and component-wise sign swapping. Therefore, we believe the difference between Glow
and RealNVP is mainly in the efficiency of approximation rather than the capability of approximation.

**Q1-5.** Is it accurate enough to directly say Glow is universal (line 191) since the family of maps it uses is limited?

**A1-5.** We agree with your concern for confusion and would like to add to line 191 that the result may not immediately
apply to the typical Glow models for image data that use the 1x1 invertible convolution layers and convolutional neural
networks for the coupling layers. Our explanation presumed a situation where Glow (or other coupling flows) is applied
to non-image data (e.g., [11, 13] in the paper). In this case, the 1x1 invertible convolution layers correspond to the
general linear group, and our results apply. Extending our results to the case of image data is future work.

**To Reviewer #2**

**Q2-1.** Ideas for improving the presentation of lines 212–235.

**A2-1.** Thank you for your suggestions for improving the manuscript. We will add such explanations accordingly.

**Q2-2.** Bigoni et al, "Greedy inference with layers of lazy maps" may be related.

**A2-2.** Thank you for pointing out the connection. We will introduce the paper in Section 5.1 as existing work that
proposed a distributionally universal class of CF-INNs (equipped with a KL-divergence approximation error bound).

**To Reviewer #3**

**Q3-1.** The results here are not quantitative. For another type of INNs (residual-flow based ones), some lower bounds
on the number of layers required for distribution approximation are known (Kong and Chaudhuri, 2020).

**A3-1.** We believe that establishing the universality of a model class remains important as the first step toward under-
standing the representation power because the question of efficiency only makes sense when the model has universality.
Nonetheless, we agree that quantitative evaluation is important, and we will investigate the question further in future
work. Our results can provide a simple route to confirming the universality not only for the existing coupling-based
flow layers, but also for those to be designed in the future for improved efficiency.

**To Reviewer #4**

**Q4-1.** Only limited architecture is analyzed.

**A4-1.** As other reviewers pointed out (e.g., Reviewer #2), the architecture of invertible neural networks based on affine
coupling flows (ACFs) is widely adopted in practice, hence we believe the analysis is highly relevant to the community
(please also see lines 19–20 of the paper and the references therein). Furthermore, the ACFs analyzed in our paper are
often special cases of more sophisticated flow layer designs, thus the result readily extends to such other architectures
as explained in lines 186–192 of the paper.

**Q4-2.** No experimental results are presented to demonstrate how the theory works on existing algorithms.

**A4-2.** We believe the present theoretical results have high importance by themselves as they tackle the long-standing
question of the universality of coupling-based invertible neural networks. The literature on the empirical evaluation of
the coupling-flow based INNs is relatively rich whereas these rigorous theoretical results have been missing ingredients,
as Reviewer #2 also pointed out.

[Meta-Review · NeurIPS 2020]

The paper received a positive feedback from the four reviewers. The reviewers have raised a few concerns, which have been addressed in the rebuttal. The area chair agrees with the reviewer's asssessment and follows their recommendation.